# Selective depletion of metastatic stem cells as therapy for human colorectal cancer

María Virtudes Céspedes[1,2,†], Ugutz Unzueta[1,2,†], Anna Aviñó[2,3], Alberto Gallardo[2,4], Patricia Álamo[1,2], Rita Sala[1,2], Alejandro Sánchez-Chardi[5], Isolda Casanova[1,2], María Antònia Mangues[1,2,6], Antonio Lopez-Pousa[2,7], Ramón Eritja[2,3] (iD), Antonio Villaverde[2,8,9,*] (iD), Esther Vázquez[2,8,9,‡] & Ramón Mangues[1,2,‡,**] (iD)

## Abstract

Selective elimination of metastatic stem cells (MetSCs) promises to block metastatic dissemination. Colorectal cancer (CRC) cells overexpressing CXCR4 display trafficking functions and metastasis-initiating capacity. We assessed the antimetastatic activity of a nanoconjugate (T22-GFP-H6-FdU) that selectively delivers Floxuridine to CXCR4+ cells. In contrast to free oligo-FdU, intravenous T22-GFP-H6-FdU selectively accumulates and internalizes in CXCR4+ cancer cells, triggering DNA damage and apoptosis, which leads to their selective elimination and to reduced tumor re-initiation capacity. Repeated T22-GFP-H6-FdU administration in cell line and patient-derived CRC models blocks intravasation and completely prevents metastases development in 38–83% of mice, while showing CXCR4 expression-dependent and site-dependent reduction in foci number and size in liver, peritoneal, or lung metastases in the rest of mice, compared to free oligo-FdU. T22-GFP-H6-FdU induces also higher regression of established metastases than free oligo-FdU, with negligible distribution or toxicity in normal tissues. This targeted drug delivery approach yields potent antimetastatic effect, through selective depletion of metastatic CXCR4+ cancer cells, and validates metastatic stem cells (MetSCs) as targets for clinical therapy.

**Keywords** colorectal cancer; CXCR4 receptor; metastatic stem cells; protein nanoconjugate; targeted drug delivery
**Subject Categories** Cancer; Digestive System; Stem Cells

## Introduction

Control of metastatic spread remains an unmet medical need. In colorectal cancer (CRC), as in other tumor types, adjuvant therapy controls metastases and prolongs survival at the expense of high toxicity; however, metastases remain the primary cause of death (Schrag, 2004; Mehlen & Puisieux, 2006; Spano *et al*, 2012; Riihimäki *et al*, 2016). There is an urgent need to develop less toxic and more effective antimetastatic agents. To achieve this goal, preclinical and clinical drug development should shift its focus from primary tumor to metastasis control, using metastatic cancer models and evaluating promising drugs in patients with limited or non-metastatic disease (Steeg & Theodorescu, 2008; Steeg, 2016). This is relevant because metastases differ from primary tumors in their mutational or gene expression profiles (Rhodes & Chinnaiyan, 2005; Vignot *et al*, 2015) and response to drugs (Takebayashi *et al*, 2013; Chen *et al*, 2015).

Metastatic stem cells (MetSCs) are a subset of cancer stem cells (CSCs) that, in addition to self-renewal and differentiation capacities, have trafficking functions (Steeg, 2016; Brabletz *et al*, 2005; Sleeman & Steeg, 2010; Oskarsson *et al*, 2014). In CRC, CXCR4 receptor enhances metastatic dissemination and confers poor patient prognosis (Kim *et al*, 2005, 2006; Schimanski *et al*, 2005), a finding similar to other cancers (Müller *et al*, 2001; Balkwill, 2004; Kucia *et al*, 2005; Schimanski *et al*, 2006; Hermann *et al*, 2007; Sun *et al*, 2010). Moreover, CXCR4-overexpressing (CXCR4+) cells have metastasis-initiating capacity (MICs) in CRC (Croker & Allan, 2008; Zhang *et al*, 2012), whereas CXCR4 RNAi-mediated downregulation or blockade of membrane localization inhibits hepatic and lung metastases (Murakami *et al*, 2013; Wang *et al*, 2014). These

1  Institut d'Investigacions Biomèdiques Sant Pau, Hospital de Santa Creu i Sant Pau, Barcelona, Spain
2  CIBER de Bioingeniería, Biomateriales y Nanomedicina (CIBER-BBN), Barcelona, Spain
3  Institute for Advanced Chemistry of Catalonia (IQAC), CSIC, Barcelona, Spain
4  Department of Pathology, Hospital de la Santa Creu i Sant Pau, Barcelona, Spain
5  Servei de Microscòpia, Universitat Autònoma de Barcelona, Barcelona, Spain
6  Department of Pharmacy, Hospital de la Santa Creu i Sant Pau, Barcelona, Spain
7  Department of Medical Oncology, Hospital de la Santa Creu i Sant Pau, Barcelona, Spain
8  Institut de Biotecnologia i de Biomedicina, Universitat Autònoma de Barcelona, Barcelona, Spain
9  Departament de Genètica i de Microbiologia, Universitat Autònoma de Barcelona, Barcelona, Spain
   *Corresponding author. Tel: +34 935813086; E-mail: antoni.villaverde@uab.cat
   **Corresponding author. Tel: +34 935537918; E-mail: rmangues@santpau.cat
   †These authors contributed equally to this work
   ‡These authors contributed equally to this work as senior authors

findings support a MetSC function for CXCR4[+] CRC cells. Nevertheless, the formal proof for MetSCs clinical relevance will only come by demonstrating that their selective targeting and elimination leads to antimetastatic effect.

Nanomedicine pursues targeted drug delivery, which aims at increasing anticancer effect while reducing toxicity (Das *et al*, 2009). We here use targeted drug delivery to CXCR4[+] MetSCs in an attempt to achieve their selective elimination. We produced the drug nanoconjugate T22-GFP-H6-FdU by covalently binding a protein nanoparticle, which selectively targets CXCR4[+] cancer cells (Unzueta *et al*, 2012a; Céspedes *et al*, 2016) to Floxuridine (FdU), a cytotoxic drug used to treat CRC liver metastases (Shi *et al*, 2015). We here demonstrate selective T22-GFP-H6-FdU biodistribution to tumor and metastatic foci in cell line- and patient-derived CRC models. We also observed its internalization and selective FdU delivery in CXCR4[+] MetSCs, leading to their depletion. After repeated T22-GFP-H6-FdU administration, and in contrast to free oligo-FdU, we achieved highly significant activity in the prevention and regression of metastases in the absence of toxicity, supporting the clinical relevance of developing drugs that selectively target MetSCs to achieve metastasis control.

# Results

### Development of T22-GFP-H6-FdU, a nanoconjugate that targets CXCR4[+] CRC cells

The previous demonstration of MIC capacity for CXCR4-overexpressing (CXCR4[+]) CRC cells (Croker & Allan, 2008; Zhang *et al*, 2012), and its inhibition by CXCR4 downregulation (Murakami *et al*, 2013; Wang *et al*, 2014), identifies these cells as MetSCs (Oskarsson *et al*, 2014). On this basis, we generated a CXCR4-targeted nanoconjugate to evaluate its capacity to achieve antimetastatic effect by selectively eliminating CXCR4[+] CRC cells. The structure and physico-chemical characterization of this new T22-GFP-H6-FdU nanoconjugate are described in Fig 1A–C, and Appendix Figs S1 and S2, which contains T22 (a ligand that targets the CXCR4 receptor), a green fluorescent

protein (allowing its *in vivo* monitoring) and oligo-FdU, an oligonucleotide of a drug active against CRC (Shi *et al*, 2015), which allows to load a high number of drug molecules into the nanoconjugate.

T22-GFP-H6-FdU was synthesized by functionalizing oligo-FdU with thiol (Fig 1C and Appendix Fig S1A), which was subsequently conjugated to the previously described T22-GFP-H6 protein nanoparticle (Unzueta *et al*, 2012a) once bound to a chemical linker (Fig 1C).

We physico-chemically characterized the HS-oligo-FdU. The functionalized pentamer FdU-HEG-SH was quantified by absorption at 260 nm and confirmed by MALDI mass spectrometry (MALDI-TOF), yielding a MW of 1,976.2, being the expected MW 1,974.0. The control pentanucleotide (free oligo-FdU) characterized by mass spectrometry (MALDI-TOF) yield a MW of 1,476.5, being the expected MW: 1,478.1. The analysis of the conjugation products was performed by MALDI-TOF spectra identifying the peaks corresponding to one or two molecules of pentaoligonucleotides of FdU bound to the nanoparticle with the MW indicated in Appendix Figs S1 and S2. The T22-GFP-H6-FdU size was determined by dynamic light scattering, being 14.6 + 0.14, as compared to 13.4 + 0.11 for the control T22-GFP-H6 nanoparticle, a size consistent with that determined by transmission electron microscopy.

This product had an approximate FdU/nanoparticle (DNR) ratio of 20 (Appendix Fig S2), and maintained its capacity for self-assembling (Unzueta *et al*, 2012a; Rueda *et al*, 2015; Appendix Fig S2D). The determined size was higher than the renal filtration cutoff (6–7 nm) ensuring a high re-circulation time in blood, a requirement for effective targeted drug delivery (Unzueta *et al*, 2012b, 2015).

### T22-GFP-H6-FdU selectively internalizes and kills CXCR4[+] CRC cells *in vitro*

Following, we used the human SW1417 CRC cell line to assess if the loaded oligo-FdU conferred cytotoxic activity to the nanoparticle while maintaining its CXCR4 targeting capacity, provided that drug conjugation can alter protein conformation and function (Goswami *et al*, 2013). We first determined that this cell line constitutively

**Figure 1.  T22-GFP-H6-FdU nanoconjugate synthesis and selective internalization and killing of CXCR4[+] CRC cells *in vitro*.**

A   The nanoconjugate contains a fusion protein [T22-GFP-H6—composed of the peptide T22 as a CXCR4 ligand, a green fluorescent protein and a histidine tail—bound to the payload drug (Unzueta *et al*, 2012a)].

B   Three to four pentameric oligonucleotides (approximately 20 molecules) of the antitumor drug 5-Fluoro-2′-deoxyuridine (FdU), named oligo-FdU, are conjugated to the T22-GFP-H6 targeting vector using a linker.

C   T22-GFP-H6-FdU chemical synthesis: T22-GFP-H6 is first covalently bound to the 6-Maleimidohexanoic acid N-hydroxysuccinimide ester linker through its amino groups in the external lysines (Hermanson, 2013). The thiol-functionalized oligo-FdU (oligo-(FdU)5-SH; see Appendix Fig S1) is then reacted with T22-GFP-H6 functionalized with maleimide (Michael reaction).

D   High and constitutive expression of CXCR4 in the membrane of SW1417 CRC cells as measured by flow cytometry.

E   Lack of human SDF-1α release from cultured SW1417 CRC cells, as measured by ELISA, whereas human control 1BR3.G fibroblasts express high SDF-1α levels, after 48 or 72 h of growth in culture (mean ± s.e.m., *N* = 2 experiment in duplicate).

F   Nanoconjugate internalization in CXCR4-overexpressing (CXCR4[+]) SW1417 CRC cells after 1-h exposure at 1 μM, as measured by fluorescence emission using flow cytometry (mean ± s.e.m., *N* = 3 experiments in duplicate). Significant difference at **$P$ = 0.002 between the T22-GFP-H6-FdU and the T22-GFP-H6-FdU + AMD3100 groups, Mann–Whitney *U*-test.

G   Intracellular trafficking of T22-GFP-H6-FdU in CXCR4[+] SW1417 cells by confocal microscopy after exposure at 1 μM for 24 h. The green staining corresponds to GFP-containing nanoconjugates, and the red staining corresponds to plasma cell membranes stained with a red dye (CellMask™), whereas cell nucleus was stained in blue with Hoechst. The insets show detail of the intracellular localization of nanostructured, fluorescent entities, in an isosurface representation within a three-dimensional volumetric *x-y-z* data field.

H   Linearized T22-GFP-H6-FdU dose–response trend line representation compared with unconjugated free oligo-FdU exposure. Antitumor effect was measured as CXCR4[+] SW1417 cell viability by MTT after 72-h exposure at the described concentrations (mean ± s.e.m., *N* = 3 experiments in duplicate).

I   Reduction in cell viability determined by optical microscope images of SW1417 cells exposed to 1 μM T22-GFP-H6-FdU for 72 h, as compared to T22-GFP-H6 or free oligo-FdU (*N* = 3 experiments in duplicate; Scale bar, 100 μm).

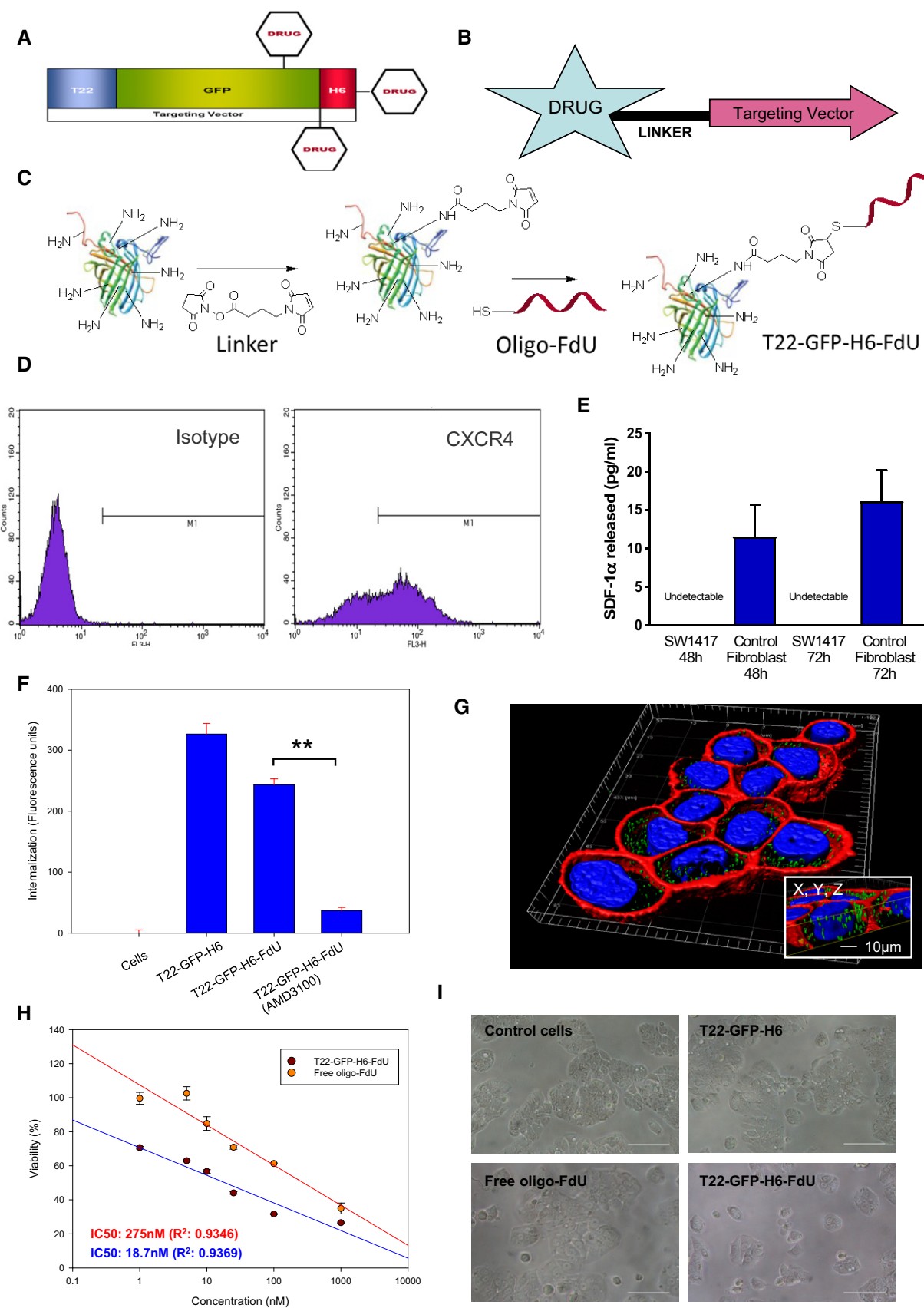

**Figure 1.**

expresses membrane CXCR4 (Fig 1D) while lacking SDF-1α expression (Fig 1E). Then, we demonstrated T22-GFP-H6-FdU capacity to internalize in CXCR4$^+$ SW1417, as measured by fluorescence emission using flow cytometry (Fig 1F), and to accumulate and traffic into its cytosol as observed by confocal microscopy (Fig 1G). The nanoconjugate maintains also its dependence on CXCR4 for internalization, since AMD3100, a CXCR4 antagonist, was able to downregulate CXCR4 receptor in the membrane and completely blocked nanoconjugate internalization (Fig 1F). In addition, T22-GFP-H6-FdU induced significantly higher cytotoxicity than free oligo-FdU in the same cells, as measured by cell viability (Fig 1H) or phase-contrast microscopy (Fig 1I). We confirmed CXCR4-dependent nanoconjugate internalization and higher cytotoxicity than free oligo-FdU in human CXCR4$^+$ HeLa cells (Appendix Fig S3A–D).

### T22-GFP-H6-FdU selectively targets CXCR4$^+$ CRC cells *in vivo*

Once CXCR4-dependence for T22-GFP-H6-FdU *in vitro* activity was established, we investigated whether the nanoconjugate could achieve targeted drug delivery after its intravenous administration in the subcutaneous (SC) CXCR4$^+$ SW1417 CRC model. We assayed its selectivity and CXCR4 dependence regarding tumor tissue uptake, internalization in CXCR4-overexpressing MetSCs (target cells), intracellular release of the cytotoxic drug FdU, and selective CXCR4$^+$ MetSC killing (Fig 2A).

T22-GFP-H6-FdU showed selective tumor uptake, as measured by fluorescence emission, 5 h after the injection of a 100 μg dose in mice (Fig 2B) as previously demonstrated for T22-GFP-H6 (Céspedes *et al*, 2016). Moreover, T22-GFP-H6-FdU selectively internalized into CXCR4$^+$ tumor cells as determined by their co-localization (merged yellow color) in the cell membrane, using dual anti-GFP and anti-CXCR4 immunofluorescence, as well as the detection of released nanoconjugate into the CXCR4$^+$ cell cytosol (green dots; Fig 2C). In addition, administering the CXCR4 antagonist AMD3100 to mice prior to the nanoconjugate completely blocked its tumor uptake (Fig 2D) as well as its internalization in CXCR4$^+$ cancer cells (Fig 2E and F). Therefore, the nanoconjugate achieves not only selective tumor biodistribution, but also its specific internalization into target CXCR4$^+$ cancer cells, in a CXCR4-dependent manner.

### T22-GFP-H6-FdU achieves targeted drug delivery leading to selective depletion of CXCR4$^+$ cancer cells in CRC tumors

We next used the same SC SW1417 CRC model to assess if the selective internalization into the cytosol of CXCR4$^+$ target cancer cells achieved by the nanoconjugate led to selective FdU delivery. We also evaluated whether the delivered FdU could induce DNA damage and caspase-3-dependent cell death, triggering the specific elimination of CXCR4$^+$ tumor cells. To that aim, we used γ-H2AX IHC to measure the generation of DNA double-strand breaks (DSBs), since they mediate FdU antitumor activity (Longley *et al*, 2003). Five hours after T22-GFP-H6-FdU treatment, the number of cells containing DSBs foci in tumors (22.8 ± 1.4) was significantly higher ($P = 0.02$) than after free oligo-FdU treatment (13.4 ± 0.7), whereas cells containing DSBs in control T22-GFP-H6 or Buffer-treated tumors were barely detectable (Fig 3A and B).

T22-GFP-H6-FdU induction of DSBs indicated its capacity to release FdU in target cells to reach the nucleus and incorporate into DNA to induce DNA damage. In addition, the number of cleaved caspase-3-positive cells signaling for apoptosis (IHC measured using anticleaved caspase-3 antibody) 5 h after T22-GFP-H6-FdU treatment (10.1 ± 1.0) was significantly higher ($P = 0.03$) than after free oligo-FdU (5.2 ± 0.9) treatment (Fig 3A and B). Moreover, increased DSB-positive cells led to higher antitumor activity, since the number of cell dead bodies, measured by Hoechst staining, which identify nuclear condensation or defragmentation, in tumor tissue 24 h after T22-GFP-H6-FdU injection was significantly ($P = 0.03$) higher (13.9 ± 0.5) than free oligo-FdU (7.1 ± 0.6), T22-GFP-H6 (3.0 ± 0.3), or Buffer (1.9 ± 0.4) treatment (Fig 3A and B).

Following, we analyzed the fraction of CXCR4$^+$ cancer cells (CXCR4$^+$ CCF) remaining in tumor tissue, along time, after a single 100 μg T22-GFP-H6-FdU dose, as compared to free oligo-FdU, using the SC CXCR4$^+$ SW1417 CRC model in NOD/SCID mice. Before treatment, both groups showed a similar CXCR4$^+$ CCF in tumor tissue (Fig 4A and B); however, after T22-GFP-H6-FdU treatment, the CXCR4$^+$ CCF was reduced at 24 h and reached its valley at 48 h

**Figure 2. Selective biodistribution and receptor-dependent uptake of T22-GFP-H6-FdU in CXCR4$^+$ cells *in vivo*.**

A  Approach to achieve targeted drug delivery and selective killing of metastatic stem cells: CXCR4-nanoconjugate interaction triggers CXCR4-mediated internalization in MetSCs, in primary tumors and metastatic foci, followed by FdU release to the cytosol and diffusion to the nucleus to induce double-strand breaks leading to selective killing of CXCR4$^+$ cells.

B  Selective T22-GFP-H6-FdU nanoconjugate biodistribution in subcutaneous CXCR4$^+$ SW1417 CRC tumor tissue 5 h after a 100 μg single intravenous dose, as measured by fluorescence emission using IVIS Spectrum 200 ($N = 5$/group). Biodistribution is similar to that achieved by the T22-GFP-H6 targeting vector and undetectable after Buffer or free oligo-FdU treatment ($N = 5$ mice/group).

C  Co-localization (yellow merged) of the T22-GFP-H6-FdU (green) and the CXCR4 receptor (red) and release of T22-GFP-H6-FdU into the cytosol in CXCR4$^+$ tumor cells 5 h after a 100 μg dose of nanoconjugate, as measured by dual anti-GFP/anti-CXCR4 immunofluorescence (IF). DAPI (blue nuclear staining). Fluorescence emission was measured in the green and red channels using the ImageJ software and expressed as mean area (A) ± s.e.m ($μm^2$) ($N = 10$, 2 tumor fields × 5 mice; 200×). Note the significant ($P = 0.003$) increase in the area occupied by the green dots (nanoconjugate released to the cell cytosol) in T22-GFP-H6-FdU-treated tumors, compared to free oligo-FdU-treated control tissues. Scale bar, 50 μm.

D  Administration of the CXCR4 antagonist AMD3100 completely blocks T22-GFP-H6-FdU tumor biodistribution, as measured by fluorescence emission. Fluorescence is not detected in Buffer or free oligo-FdU controls ($N = 5$ tumor fields/group).

E  The uptake of T22-GFP-H6-FdU observed in CXCR4$^+$ SW1417 tumor tissues is almost completely blocked by prior AMD3100 administration, as quantified using the anti-GFP IHC H-score (mean ± s.e.m., $N = 5$ tumor fields/group). Comparison of T22-GFP-H6 uptake between groups: (B: Buffer; F: free oligo-FdU; T-F: T22-GFP-H6-FdU; T-F+A: T22-GFP-H6-FdU+AMD3100). $P$-values for statistical differences B vs. T-F, **$P = 0.000$; F vs. T-F, **$P = 0.000$; T-F vs. TFA, **$P = 0.004$. Mann–Whitney $U$-test.

F  Representative images of T22-GFP-H6-FdU uptake and AMD3100 competition by anti-GFP immunostaining, which quantitation is reported in panel (E). Scale bar, 50 μm.

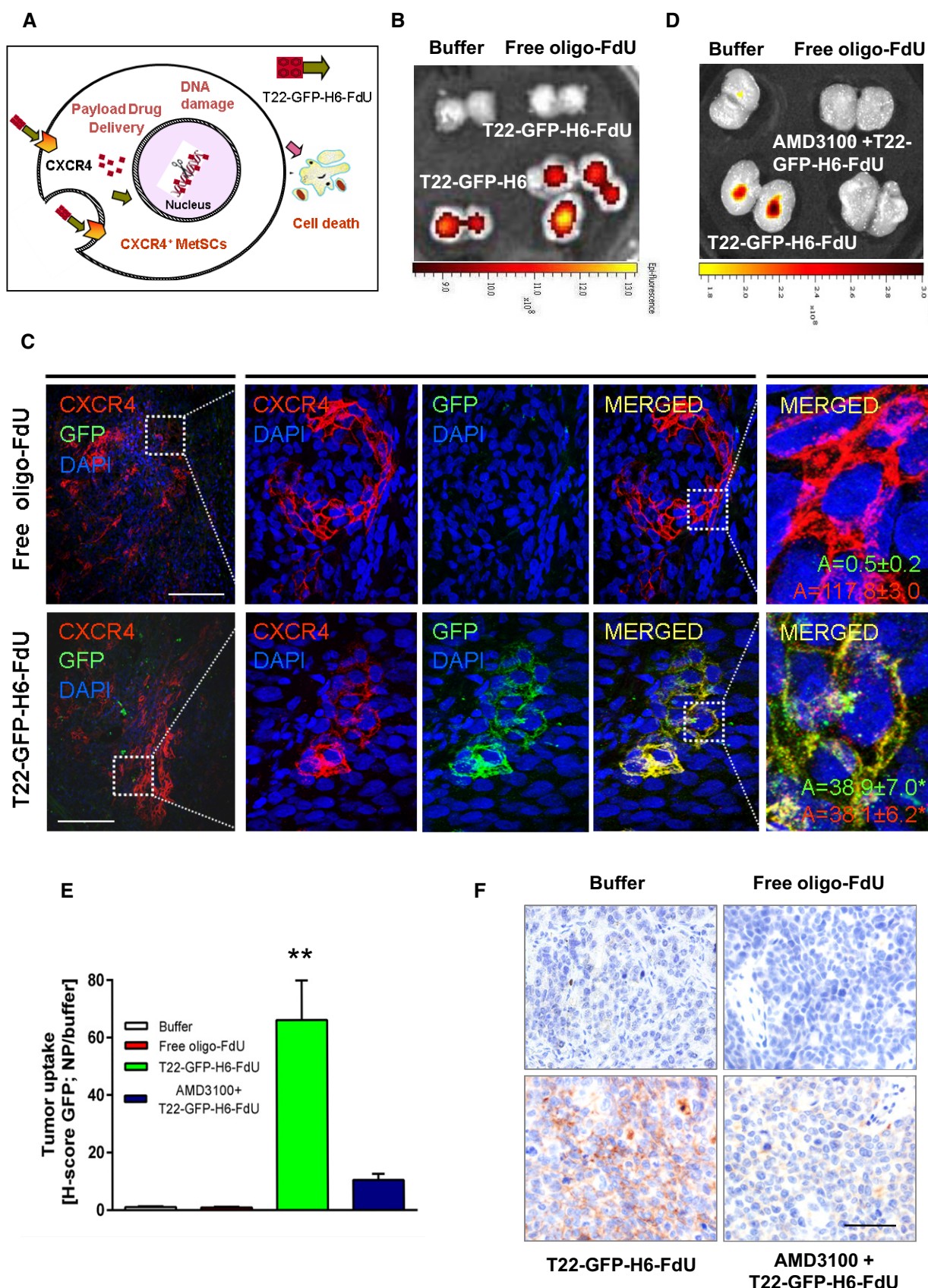

**Figure 2.**

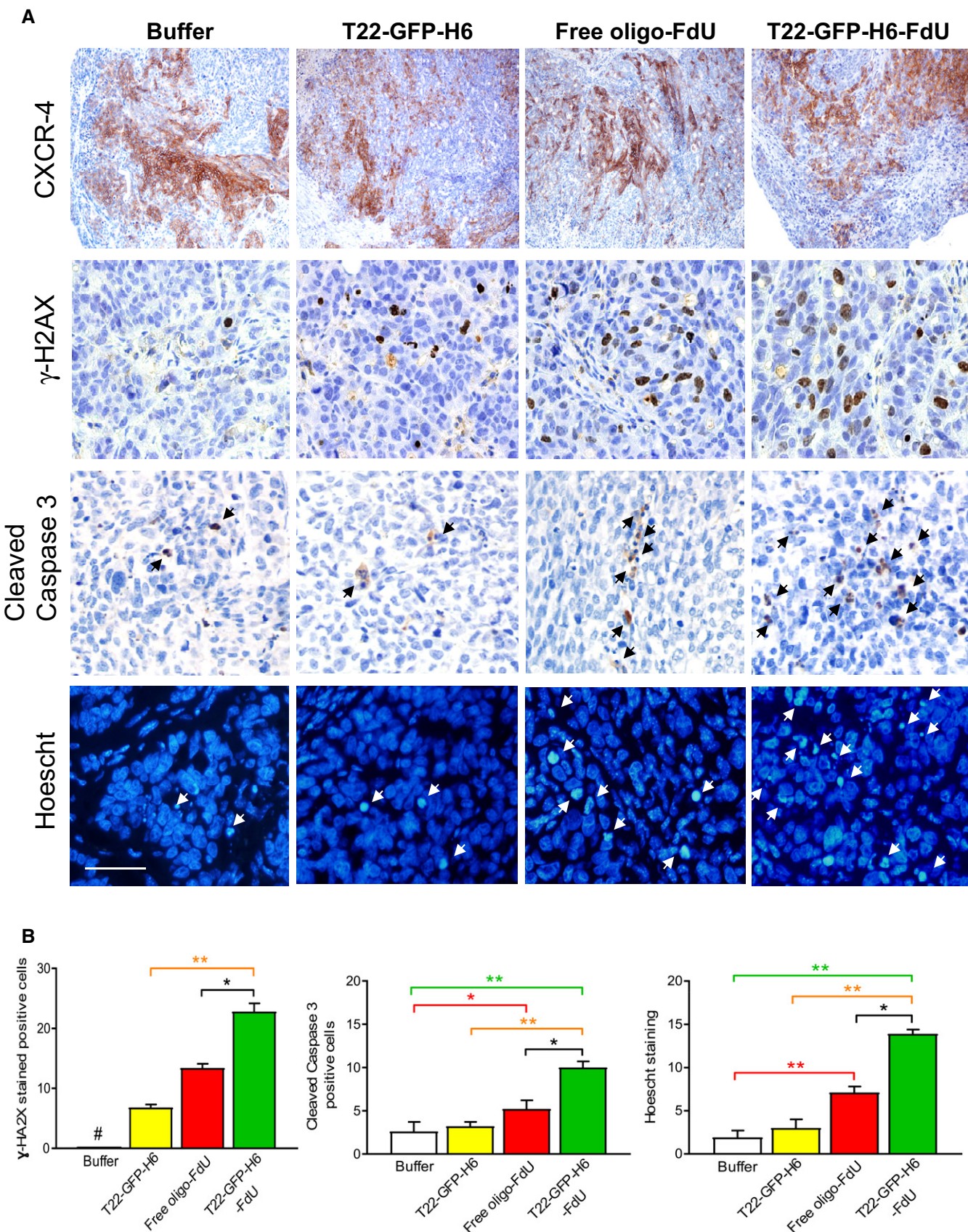

**Figure 3.**

**Figure 3.   T22-GFP-H6-FdU-induced depletion of CXCR4-overexpressing cancer cells in tumor tissue.**

A   Representative images of CXCR4 overexpression in subcutaneous tumor tissue, showing similar CXCR4 levels among compared groups ($N$ = 5/group; Buffer, T22-GFP-H6-FdU, T22-GFP-H6, and free oligo-FdU) before treatment (upper panels). Representative images of DNA double-strand break induction and caspase-3 activation (measured with anti-$\gamma$-H2AX or anticleaved caspase-3 by IHC) 5 h post-administration (middle panels). Apoptotic induction (Hoechst staining, 24 h post-administration, lower panels). Note the higher number of cells positive for DSBs, caspase-3 activation, and apoptosis induction in the T22-GFP-H6-FdU as compared to free oligo-FdU. Black or white arrows indicate dead cells. Scale bar, 50 $\mu$m.

B   Quantitation of the number of cells containing DSBs or active caspase-3 in IHC-stained tumor sections 5 h post-treatment and the number of condensated or disaggregated nuclei (by Hoechst staining) 24 h post-treatment in tumor sections of 10 high-power fields (400× magnification) using the Cell^D software ($N$ = 50; 10 tumor fields/mice; 5 mice/group). Data expressed as mean ± s.e.m. Parameter comparison between groups: (B: Buffer; T: T22-GFP-H6; F: free oligo-FdU; T-F: T22-GFP-H6-FdU). *P*-values for statistical differences: $\gamma$-H2AX staining quantitation: B vs. T, $^{\#}P$ = 0.001; B vs. F, $^{\#}P$ = 0.000; B vs. T-F, $^{\#}P$ = 0.000; T vs. T-F, **$P$ = 0.001; F vs. T-F, *$P$ = 0.02. Cleaved caspase-3 quantitation: B vs. F, *$P$ = 0.034; B vs. T-F, **$P$ = 0.009; T vs. T-F, **$P$ = 0.003; F vs. T-F, *$P$ = 0.012. Hoechst staining quantitation: B vs. F, **$P$ = 0.01; B vs. T-F, **$P$ = 0.001; T vs. T-F, **$P$ = 0.000; F vs. T-F, *$P$ = 0.032. Mann Whitney *U*-test.

(Fig 4A and B). In contrast, the CXCR4$^+$ CCF in tumor tissue after an equimolecular dose of free oligo-FdU remained similar to its basal level along time. Taken together, these results indicate that T22-GFP-H6-FdU achieves selective biodistribution to tumor tissue and FdU delivery to target CXCR4$^+$ cancer cells, as indicated by an enhancement in DNA damage and apoptotic tumor cell death, which triggers selective elimination of CXCR4$^+$ cancer cells *in vivo*, achieving, therefore, targeted FdU delivery to target cancer cells.

**Transient target cell elimination and definition of a dose interval for repeated T22-GFP-H6-FdU injection**

Despite T22-GFP-H6-FdU achieved selective depletion of CXCR4$^+$ target cells in tumor tissue observed 48 h after its administration, we found this effect to be transient, since 72 h post-injection CXCR4$^+$ cancer cell fraction in tumor tissue grew back, nevertheless, to reach a level lower than that basal before therapy (Fig 4A and B). In contrast, the CXCR4$^+$ CCF in tumor tissue after free oligo-FdU therapy was maintained over time, remaining similar at 24, 48, or 72 h after treatment as before therapy (Fig 4A and B). Therefore, in contrast to T22-GFP-H6-FdU effect, cancer killing by free oligo-FdU did not show selectivity toward CXCR4$^+$ cancer cells. Based on these results, and in order to evaluate T22-GFP-H6-FdU antimetastatic effect, we defined a 72 h (3 days) dose interval as optimal for its administration in a repeated dose schedule. We expected this regime to maintain sufficiently low the fraction of CXCR4$^+$ cancer cells remaining in primary tumors and metastatic foci, along the treatment period, as to efficiently block metastasis and/or foci growth, provided that CXCR4$^+$ cancer cells act as MetSCs.

**T22-GFP-H6-FdU-treated tumors reduce their spheroid formation and tumor re-initiation capacities**

We next used the CXCR4$^+$ luciferase$^+$ SW1417 SC CRC model to assess if the selective CXCR4$^+$ cancer cell killing induced by T22-GFP-H6-FdU treatment *in vivo* was capable of blocking spheroid formation *in vitro*. Thus, we cultured $1 \times 10^6$ disaggregated cells in stem cell- conditioned media and low-adhesion plates that were obtained from CXCR4$^+$ luciferase$^+$ SW1417 subcutaneous tumors, 24 h after 100 $\mu$g T22-GFP-H6-FdU i.v. doses, for 2 consecutive days, and observed a reduction in spheroid formation (Fig 4C), as compared to cells obtained after an equimolar free-FdU or Buffer treatment. The bioluminescence intensity emitted by the spheroids generated after T22-GFP-H6-FdU treatment ($9.1 \times 10^7 \pm 3.2 \times 10^7$) was significantly ($P$ = 0.02) reduced as compared to free oligo-FdU

($19.0 \times 10^7 \pm 0.38 \times 10^7$) or Buffer ($40.0 \times 10^7 \pm 1.9 \times 10^7$) treatment (Fig 4D). Similarly, culture of $1 \times 10^6$ disaggregated cells obtained from patient-derived CXCR4$^+$ M5 subcutaneous tumors, treated with the same T22-GFP-H6-FdU dosage, leads to a significant ($P$ = 0.001) reduction in the number of formed spheroids ($19.1 \pm 1.2$), as compared to free oligo-FdU-treated ($46.3 \pm 3.1$) or Buffer-treated ($73.1 \pm 7.0$) mice (Fig 5A and B).

In addition, the inoculation of $5 \times 10^6$ cells, subcutaneously in recipient NSG mice, derived from disaggregated tumor cells obtained from CXCR4$^+$ M5 SC tumors after the administration of 100 $\mu$g T22-GFP-H6-FdU intravenous doses, for 2 consecutive days, leads to a reduction in tumor re-initiation (as measured as diminished number and size of tumors) 10 days after the end of treatment, as compared to free oligo-FdU-treated or Buffer-treated mice (Fig 5C and D). Thus, in both, the SW1417 and the M5 CRC models CXCR4$^+$ cancer cells behave as cancer stem cells since their selective elimination reduces their tumor re-initiation capacity.

**T22-GFP-H6-FdU-induced blockade of tumor emboli intravasation**

Following, we assessed if T22-GFP-H6-FdU treatment of patient-derived CXCR4$^+$ M5 orthotopic tumors blocked dissemination from the primary tumors at an early time. To this aim, 7 days after implantation of two million M5 tumor cells in the mouse cecum, we administered 100 $\mu$g T22-GFP-H6-FdU intravenous doses, for 2 consecutive days, and 24 h later sacrificed the mice and proceed to H&E staining of samples from tumors and peri-tumoral areas. We observed that T22-GFP-H6-FdU administration induced a significant ($P$ = 0.016) reduction in the number of intravasated tumor emboli within the vessels of the peri-tumoral area ($1.6 \pm 0.4$), which were microscopically detected, in comparison with free oligo-FdU-treated ($5.4 \pm 1.3$) or Buffer-treated ($5.1 \pm 2.0$) tumors (Fig 5E and F). Moreover, T22-GFP-H6-FdU treatment reduced also significantly ($P$ = 0.027) the H-score (percent and intensity of IHC stained and normalized by foci area) for CXCR4 expression in peri-tumoral intravasated tumor emboli ($0.017 \pm 0.012$), as compared to free oligo-FdU-treated ($0.043 \pm 0.010$) or Buffer-treated ($0.038 \pm 0.005$) tumors (Fig 5E–G). Thus, T22-GFP-H6-FdU blocks tumor emboli intravasation in the primary tumor peri-tumoral vessels.

**T22-GFP-H6-FdU induces the regression of established metastases**

We assessed T22-GFP-H6-FdU capacity to inhibit growth of established metastases, as compared to equimolecular doses of T22-GFP-

Figure 4.

**Figure 4. T22-GFP-H6-FdU-induced depletion of CXCR4-overexpressing cancer cells in tumor tissue leading to reduced spheroid formation capacity.**

A, B    T22-GFP-H6-FdU depletes CXCR4+ cancer cells from SW1417 CRC tumor tissue after a 100 μg single-dose administration. Note the reduction in CXCR4+ cell fraction in the tumor 24 h after injection, their almost complete elimination at 48 h, and the re-emergence of CXCR4+ cells 72 h post-administration, using anti-CXCR4 IHC. In contrast, the CXCR4+ cancer cell fraction (CXCR4+ CCF) in tumor tissue remains constant along time after free oligo-FdU treatment. The 3-day time-lapse for CXCR4+ tumor cell re-appearance defines the dosage interval used in a repeated dose schedule of nanoconjugate administration in the experiments to evaluate its antimetastatic effect (N = 5: 5 mice/group; 1 samples/mouse). Scale bar, 50 μm. Data expressed as mean ± s.e.m. CXCR4 H-score comparison for T22-GFP-H6-FdU(T-F)-treated tumors among time points (green line, panel A). *P*-values for statistical differences: T-F Basal vs. T-F 24 h, *P = 0.038; T-F Basal vs. T-F 48 h, **P = 0.001; T-F Basal vs. T-F 72 h, **P = 0.003; T-F 24 h vs. T-F 48 h, *P = 0.033. CXCR4 H-score comparison between T22-GFP-H6-FdU (T-F) and free oligo-FdU (F) (black line, panel A). *P*-values for statistical differences: T-F vs. F at 48 h, **P = 0.001; T-F vs. F at 72 h, *P = 0.034. Mann–Whitney *U*-test.

C, D    Significant reduction in the number of spheroid formed (C, optical microscope) and their bioluminescence emission (D, IVIS Spectrum 200), generated by 1 × 10⁶ disaggregated cells (cultured in stem cell-conditioned media and low-adhesion plates), obtained from CXCR4+ luciferase+ SW1417 subcutaneous tumors, 24 h after 100 μg T22-GFP-H6-FdU intravenous doses, for 2 consecutive days, as compared to Buffer-treated or free oligo-FdU-treated mice. (D) Quantitation of the bioluminescent signal (BLI) expressed as average radiant intensity, obtained using the IVIS spectrum 200 equipment (N = 2 plates/group). Data expressed as mean ± s.e.m. Comparison of emitted BLI between groups: (B: Buffer; F: free oligo-FdU; T-F: T22-GFP-H6-FdU). *P*-values for statistical differences: T-F vs. B, **P = 0.001 (green line, panel D); F vs. B, *P = 0.011 (red line); T-F vs. F at *P = 0.02 (black line, panel D). Mann–Whitney *U*-test.

H6 or free oligo-FdU, using an orthotopic bioluminescent CXCR4+ CRC model in *Swiss nude* mice, which generates lymph node (LN) and lung (LG) metastases (Mets), starting therapy 2 months after CRC cell implantation, given a 20 μg i.v. q3d dosage (Appendix Fig S5A). At the end of the regression of metastasis experiment, T22-GFP-H6-FdU-treated mice registered a lower number of LG Mets than free oligo-FdU, as measured by *ex vivo* bioluminescence emission (Appendix Fig S6A). This was confirmed by the finding of 3.0- and 2.9-fold reduction in total and mean LG foci number in histology sections of the T22-GFP-H6-FdU group as compared to free oligo-FdU (P = 0.04) mice (Appendix Fig S6B and C). T22-GFP-H6-FdU mice had a significantly lower number of LN Mets than Buffer-treated mice (P = 0.03); however, its effect was similar to that achieved by free oligo-FdU treatment.

## T22-GFP-H6-FdU prevents hematogenous and transcelomic metastases in the SW1417 cell line-derived CRC model

We also evaluated T22-GFP-H6-FdU capacity to prevent metastasis as compared to free oligo-FdU, by registering the percent of mice with undetectable metastases at the end of treatment (Mets-free mice) and the reduction in number and size of Mets foci in mice with detectable metastases (Mets+ mice) at the end of the experiment, using the CXCR4+ SW1417 orthotopic bioluminescent CRC model, which metastasizes to lymph nodes (LN), liver (LV), lung (LG), and peritoneum (PTN), starting treatment 1 week after CRC implantation and following a schedule of 20 μg, q3d, 12 doses (Appendix Fig S5B).

At the end of the experiment, and in contrast to findings in Buffer or oligo-FdU groups, T22-GFP-H6-FdU treatment potently prevented hematogenous (LV and LG) and transcelomic (PTN) Mets development, whereas its capacity to prevent LN Mets was low. Thus, the percent of LV, LG, and PTN Mets-free mice after Buffer treatment was 45–55 and 27–64% for oligo-FdU, whereas T22-GFP-H6-FdU treatment increased significantly (P = 0.004) to reach 83% of Mets-free mice at all sites (Table 1 and Appendix Table S1). The differences in Mets-free mice between Buffer and oligo-FdU mice were not significant.

Consistently, in Mets+ mice, T22-GFP-H6-FdU reduced the LV and LG Mets number, as measured by *ex vivo* bioluminescence compared to free oligo-FdU effect (data not shown). Moreover, a histological analysis of the foci number and size in LV, LG, and PTN Mets+ mice at the end of treatment showed that T22-GFP-H6-FdU

mice had a 7.3- and 7.0-fold reduction in the total and mean PTN foci number (P = 0.0001), and a 2.4-fold reduction in PTN foci size (P = 0.01) as compared to free oligo-FdU (Appendix Fig S8A and Table 1). Similarly, T22-GFP-H6-FdU induced a 2.7- and 5.0-fold reduction in total and mean number of LV (P = 0.001) or LG (4.5- and 3.5-fold, P = 0.006) Mets as compared to free oligo-FdU, and only a mild effect on LN Mets. Importantly, free oligo-FdU did not reduce the total or mean Mets foci number at any site (LN, LV, LG, PTN), as compared to Buffer-treated animals (Appendix Fig S8A and Table 1).

## T22-GFP-H6-FdU prevents hematogenous and transcelomic metastases in the M5 patient-derived CRC model

We, next assessed T22-GFP-H6-FdU capacity to prevent LN, LV, LG, and PTN Mets development in the CXCR4+ M5 orthotopic CRC model, which shows higher metastatic efficiency at all sites (Table 1). To that aim, we measured all parameters described above in the SW1417 model and applied a schedule of 20 μg, q3d, per seven doses, starting 1 week after tumor cell implantation (Appendix Fig S5C). T22-GFP-H6-FdU treatment potently prevented LV, LG, and PTN Mets development, whereas its capacity to prevent LN Mets was low (Table 1). Thus, the percent of LV, LG, and PTN Mets-free mice was 0% after Buffer treatment and 15–30% after free oligo-FdU treatment, whereas T22-GFP-H6-FdU treatment significantly (P = 0.05) increased this effect to reach 38–63% of LV, LG, and PTN Mets-free mice. The differences between Buffer and oligo-FdU treatment were not significant (Table 1 and Appendix Table S1).

The histological evaluation of LV, LG, and PTN foci in Mets+ mice at the end of treatment showed that T22-GFP-H6-FdU mice registered a 9.0- and 9.4-fold reduction in the total and mean LV foci number (P = 0.001), and 12.1-fold reduction in LV foci size (P = 0.007) as compared to free oligo-FdU (Fig 6A and Table 1). Similarly, T22-GFP-H6-FdU induced a 5.7- and 2.7-fold reduction in total and mean number of PTN (P = 0.022) or LG (2.4- and 2.8-fold, P = 0.003) Mets as compared to free oligo-FdU and having, and only a mild effect on LN Mets (Fig 6A and Table 1). Importantly, in contrast to T22-GFP-H6-FdU, free oligo-FdU did not reduce the total or mean Mets foci number at any site (LN, LV, LG, PTN), as compared to Buffer-treated animals (Fig 6A and Table 1).

In summary, repeated T22-GFP-H6-FdU administration potently prevented the development of hematogenous (LV and LG) and transcelomic (PTN) metastases yielding a 38–83% of Mets-free mice,

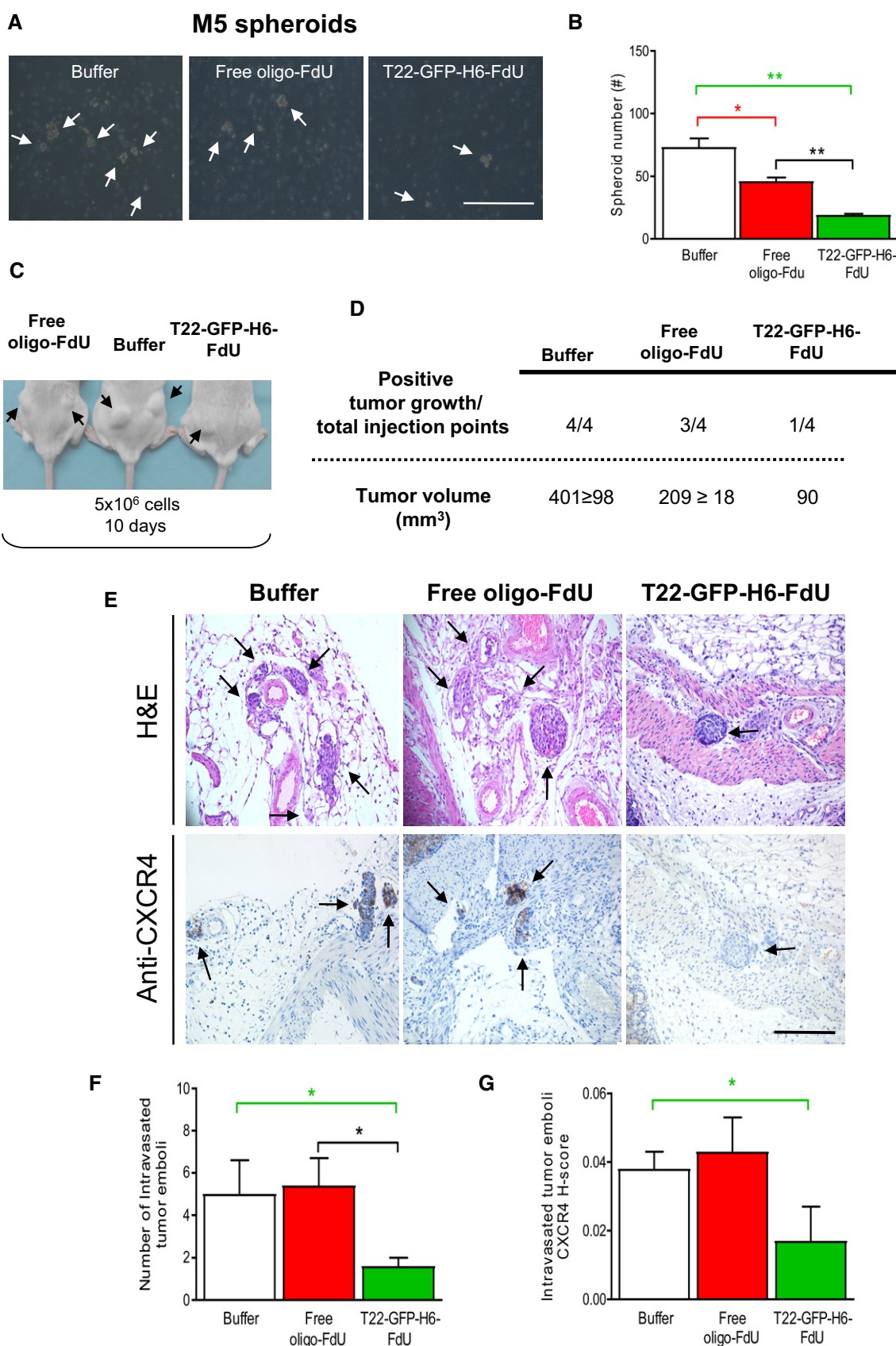

**Figure 5.**

◄

**Figure 5.  T22-GFP-H6-FdU-induced reduction in tumor re-initiation capacity and blockade of tumor emboli intravasation in the CXCR4+ patient-derived M5 model.**

A, B   Reduction in the number of formed spheroids (white arrows, optical microscope) generated by $1 \times 10^6$ disaggregated cells (cultured in stem cell-conditioned media and low-adhesion plates) obtained from CXCR4+ M5 subcutaneous tumors, 24 h after 100 µg T22-GFP-H6-FdU intravenous doses, for 2 consecutive days, as compared to Buffer-treated or free oligo-FdU-treated mice (mean ± s.e.m., N = 8; 2 mice/group; 4 plates/mouse). Scale bar, 100 µm. Comparison of spheroid formation between groups: (B: Buffer; F: free oligo-FdU; T-F: T22-GFP-H6-FdU). *P*-values for statistical differences: T-F vs. B, **P = 0.001 (green line, Panel B); F vs. B, *P = 0.012 (red line); T-F vs. F, **P = 0.001 (black line). Mann–Whitney *U*-test.

C, D   Reduction in tumor re-initiation capacity after subcutaneous inoculation of $5 \times 10^6$ cells in NSG mice (N = 4 tumors/group) derived from disaggregated tumor cells obtained from SC tumors 10 days after administration of 100 µg T22-GFP-H6-FdU intravenous doses, for 2 consecutive days, as compared to free oligo-FdU-treated or Buffer-treated mice. Recording of the number and size of positive tumors (black arrows, N = 4; 2 mice/group, 2 injection points/mouse).

E–G   (E) Representative images of tumor emboli intravasation determined by microscopic analyses of H&E-stained tumor sections (N = 5/group). (F) T22-GFP-H6-FdU-induced reduction in the number of intravasated tumor emboli (black arrows) in peri-tumoral vessels of the M5-orthotopic primary tumor (E: optical images; F: emboli number quantitation) and reduction in CXCR4 expression in these emboli (G), treated 7 days after tumor cell implantation with 100 µg T22-GFP-H6-FdU intravenous doses, for 2 consecutive days, as compared to Buffer-treated or free oligo-FdU-treated mice. Tumor emboli counting in 10 high-power field at 200× magnification in H&E-stained sections from each tumor (mean ± s.e.m., N = 5/group). Comparison of tumor emboli number between groups: (B: Buffer; F: free oligo-FdU; T-F: T22-GFP-H6-FdU). *P*-values for statistical differences: T-F vs. B, *P = 0.038 (green line), T-F vs. F, *P = 0.016 (black line). Scale bar, 100 µm. (G) CXCR4 expression per tumor emboli determined by using anti-CXCR4 IHC and calculating H-score (multiplying percent of CXCR4+ cells out of total cell number in the emboli area by their staining intensity, scoring each from 0 to 3 (where 3 is the maximal intensity) per tumor emboli area (mean ± s.e.m., N = 5 mice/group). Comparison of CXCR4 H-score between groups: (B: Buffer; T-F: T22-GFP-H6-FdU). *P*-values for statistical differences: T-F vs. B at *P = 0.027 (green line). Mann–Whitney *U*-test.

depending on the site and studied model. It also reduced significantly the number and/or size of LV, LG, and PTN foci in Mets+ mice. Nevertheless, T22-GFP-H6-FdU was unable to block LN Mets development. In contrast, free oligo-FdU did not prevent metastases at any site. In addition, T22-GFP-H6-FdU was more potent than free oligo-FdU in inducing the regression of established LG Mets. Interestingly, both T22-GFP-H6-FdU and free oligo-FdU showed a similar inhibitory effect on primary tumor growth as measured by *in vivo* bioluminescence emission along time or *ex vivo* at the end of treatment, both in the prevention or regression of metastasis experiments (Appendix Figs S6A and B, and S7A–D).

**Site-dependent CXCR4 regulation, T22-GFP-H6-FdU CXCR4+ cell targeting, and antimetastatic effect**

Based on the clear site-dependent antimetastatic potency achieved by T22-GFP-H6-FdU in the prevention of metastasis experiments (Fig 6A, Appendix Fig S8A, and Table 1), on its dependence on CXCR4 membrane expression for cell internalization (Fig 2E) and capacity to selectively kill CXCR4+ cancer cells (Fig 3A and B), we investigated if CXCR4 expression after therapy correlated with the observed antimetastatic effect at the different sites.

We observed a site-dependent reduction in CXCR4+ target cancer cell fraction (CXCR4+ CCF) in Mets foci at the end of T22-GFP-H6-FdU treatment, as detected by anti-CXCR4 IHC, (and as compared to basal levels) which correlated with the antimetastatic effect at the different sites in both SW1417 and M5 patient-derived CRC models (Fig 6B, Appendix Fig S8B, and Table 1). The LV, LG, and PTN Mets, highly sensitive to T22-GFP-H6-FdU treatment in terms of increased percent of Mets-free mice and reduction in foci number and size in Mets+ mice, reached the lowest level of CXCR4+ CCF at the end of treatment at these sites. In contrast, in both the M5 and SW1417 models we observed only a low and non-significant reduction in CXCR4+ CCF in the organs showing low sensitivity to T22-GFP-H6-FdU, such as the primary tumor or LN Mets (Fig 6B and C, and Appendix Fig S8B and C). Moreover, conversely to findings with to T22-GFP-H6-FdU, free oligo-FdU did not reduce CXCR4+ CCF at any Mets site (Fig 6A and Appendix Fig S8A). Similarly, in the regression of metastasis experiment, we observed a CXCR4+

CCF reduction in LG Mets and higher antimetastatic effect at this site than in LN Mets, which showed no reduction in CXCR4+ CCF and poor response to T22-GFP-H6-FdU therapy (Appendix Fig S6C and D and Table 1).

**Lack of T22-GFP-H6-FdU accumulation or toxicity in normal tissues**

To estimate the T22-GFP-H6-FdU therapeutic window, we analyzed its biodistribution and induction of DNA damage and apoptosis in non-tumor tissues. T22-GFP-H6-FdU injection led to highly selective tumor tissue accumulation (Fig 2B) as measured by fluorescence emission, whereas uptake in CXCR4-positive (bone marrow or spleen) or CXCR4-negative (kidney, lung, brain, heart or liver) normal tissues was undetectable, except for a transient accumulation in the liver (Fig 7A), in the same experiment. Moreover, the number of cells containing DSBs, detected by anti-γ-H2AX IHC, in normal bone marrow 5 h after T22-GFP-H6-FdU treatment (6.1 ± 1.2) was significantly lower (P = 0.047) than in free oligo-FdU-treated mice (11.4 ± 0.9; Fig 7B), whereas DSB-positive cells in normal liver or kidney were similarly low in all compound-treated groups or Buffer-treated animals. Moreover, DSBs induction did not lead to apoptosis or histological alteration in any group, since no histological alterations were detected in bone marrow, liver, or kidney 24 h post-administration (Fig 7C). Therefore, consistently with the negligible nanoconjugate distribution to normal tissues, the lack of detectable apoptosis or histological alterations in all analyzed tissues, including bone marrow or circulating blood monocytes (Appendix Fig S9), the lack of mouse body weight loss in the regression or prevention (Fig 7D–F) of metastases experiments, and the absence of any sign of clinical toxicity indicate a wide therapeutic index for T22-GFP-H6-FdU at a dosage that achieves potent antimetastatic effect.

# Discussion

The identification of CXCR4+ tumor cells as metastasis stem cells (MetSCs) (Oskarsson *et al*, 2014) in colorectal cancer (CRC; Croker

**Table 1.** T22-GFP-H6-FdU antimetastatic effect, observed in the prevention of metastasis experiments in the SW1417 and M5 CRC metastatic models, measured as percent of mice free of metastases at the end of treatment and as reduction in mean foci number and foci size in metastasis-positive mice[†].

**Prevention of metastasis protocol**

**SW1417 cell line-derived orthotopic model**

| Groups | Lymph node Mets (LNm) | | Liver Mets (LVm) | | Lung Mets (LGm) | | Peritoneal Mets (PTNm) | |
|---|---|---|---|---|---|---|---|---|
| | % Mice free of LN Mets | Foci # in Mets+ mice | % Mice free of LV Mets | Foci # in Mets+ mice | % Mice free of LG Mets | Foci # in Mets+ mice | % Mice free of PTN Mets | Foci # in Mets+ mice |
| Buffer | 0% | $3.7 \pm 0.3^a$ | 64% | $0.7 \pm 0.3^b$ | 27% | $6.6 \pm 1.5^d$ | 36% | $2.0 \pm 0.6^f$ |
| Free oligo-FdU | 0% | $3.1 \pm 0.4$ | 45% | $1.0 \pm 0.3^c$ | 45% | $4.5 \pm 1.6^e$ | 55% | $2.8 \pm 1.0^g$ |
| T22-GFP-H6-FdU | 25% | $2.0 \pm 0.4^a$ | 83% | $0.2 \pm 0.1^{b,c}$ | 83% | $1.3 \pm 0.9^{d,e}$ | 83% | $0.4 \pm 0.3^{f,g}$ |

**Metastatic Foci size[‡] ($\mu m^2 \times 10^{-3}$)**

| Groups | Lymph node Mets (LNm) | Liver Mets (LVm) | Lung Mets (LGm) | Peritoneal Mets (PTNm) |
|---|---|---|---|---|
| Buffer | $110.7 \pm 15.5^o$ | $11.2 \pm 3.4$ | $21.2 \pm 1.3^p$ | $435.7 \pm 67.2^q$ |
| Free oligo-FdU | $77.0 \pm 14.2^o$ | $9.6 \pm 2.3$ | $17.9 \pm 1.6$ | $304.8 \pm 22.3^r$ |
| T22-GFP-H6-FdU | $79.3 \pm 11.1$ | $8.7 \pm 2.6$ | $15.1 \pm 2.2^p$ | $126.6 \pm 18.7^{q,r}$ |

**M5 patient-derived orthotopic model**

| Groups | Lymph node Mets (LNm) | | Liver Mets (LVm) | | Lung Mets (LGm) | | Peritoneal Mets (PTNm) | |
|---|---|---|---|---|---|---|---|---|
| | % Mice free of LN Mets | Foci # in Mets+ mice | % Mice free of LV Mets | Foci # in Mets+ mice | % Mice free of LG Mets | Foci # in Mets+ mice | % Mice free of PTN Mets | Foci # in Mets+ mice |
| Buffer | 0% | $50.0 \pm 18.4^h$ | 0% | $10.5 \pm 2.5^i$ | 0% | $19.5 \pm 7.1^k$ | 0% | $40.0 \pm 11.4^{l,m}$ |
| Free oligo-FdU | 0% | $39.6 \pm 14.6$ | 15% | $7.5 \pm 2.7^j$ | 30% | $16.2 \pm 6.9$ | 15% | $19.7 \pm 8.4^{l,n}$ |
| T22-GFP-H6-FdU | 0% | $22.7 \pm 4.1^h$ | 63% | $0.8 \pm 0.5^{i,j}$ | 50% | $5.8 \pm 2.4^k$ | 38% | $7.2 \pm 1.8^{m,n}$ |

**Metastatic Foci size[‡] ($\mu m^2 \times 10^{-3}$)**

| Groups | Lymph node Mets (LNm) | Liver Mets (LVm) | Lung Mets (LGm) | Peritoneal Mets (PTNm) |
|---|---|---|---|---|
| Buffer | $1,749.4 \pm 434.7$ | $20.0 \pm 6.1^s$ | $10.8 \pm 1.5$ | $3,603.3 \pm 976.6$ |
| Free oligo-FdU | $1,808.4 \pm 289.2$ | $38.7 \pm 11.7^t$ | $7.7 \pm 0.9^u$ | $3,665.6 \pm 589.2$ |
| T22-GFP-H6-FdU | $1,752.2 \pm 426.3$ | $3.2 \pm 0.7^{s,t}$ | $10.2 \pm 1.4^u$ | $3,057.7 \pm 1,415.4$ |

Mean + s.e.m. metastatic foci number per mouse counted in three randomly chosen histology sections.

Free oligo-FdU: equimolecular doses of free oligo-FdU.

[†]See Appendix Table S1 for detailed data on metastasis-free mice and statistical analysis.

[‡]Mean + s.e.m. metastatic foci area ($\mu m^2$) per mouse counted in three randomly chosen histology sections.

[a]$P = 0.04$; [b]$P = 0.01$; [c]$P = 0.001$; [d]$P = 0.002$; [e]$P = 0.006$; [f]$P = 0.002$; [g]$P = 0.006$; [h]$P = 0.006$; [i]$P = 0.001$; [j]$P = 0.001$; [k]$P = 0.003$; [l]$P = 0.015$; [m]$P = 0.001$; [n]$P = 0.022$; [o]$P = 0.009$; [p]$P = 0.032$; [q]$P = 0.002$; [r]$P = 0.01$; [s]$P = 0.02$; [t]$P = 0.007$; [u]$P = 0.017$.

& Allan, 2008; Zhang *et al*, 2012; Murakami *et al*, 2013; Wang *et al*, 2014) allowed us to evaluate the clinical relevance of targeting CRC MetSCs by assessing whether their selective elimination induces antimetastatic activity. Our nanotechnology approach achieved the goal of targeted drug delivery (Das *et al*, 2009) to MetSCs by taking advantage of their membrane CXCR4 overexpression, as compared to normal tissues (Kim *et al*, 2005, 2006; Schimanski *et al*, 2005). The T22-GFP-H6-FdU nanoconjugate replicates the nanoparticle capacities for self-assembling, lack of renal filtration, high re-circulation in blood, and selective internalization in target CXCR4+ cells, which we described for T22-GFP-H6 (Unzueta *et al*, 2012a,b, 2015; Rueda *et al*, 2015; Céspedes *et al*, 2016) adding the ability to

transport and intracellularly release FdU, which induces an increase in genotoxic damage and apoptosis, leading to selective CXCR4+ cancer cell elimination as well as to a reduction in tumor re-initiation capacity.

The nanoconjugate achieved potent and site-dependent metastasis prevention, since its administration generated a significantly higher percent of Mets-free mice at the end of treatment, and a significant reduction in metastatic foci number and size. These effects associated with a reduction in CXCR4+ target cancer cell fraction in tumor tissue, being both the antimetastatic effect and the reduction in CXCR4+ CCF highly significant in LV, LG, and PTN Mets, whereas they were non-significant in primary tumor or LN

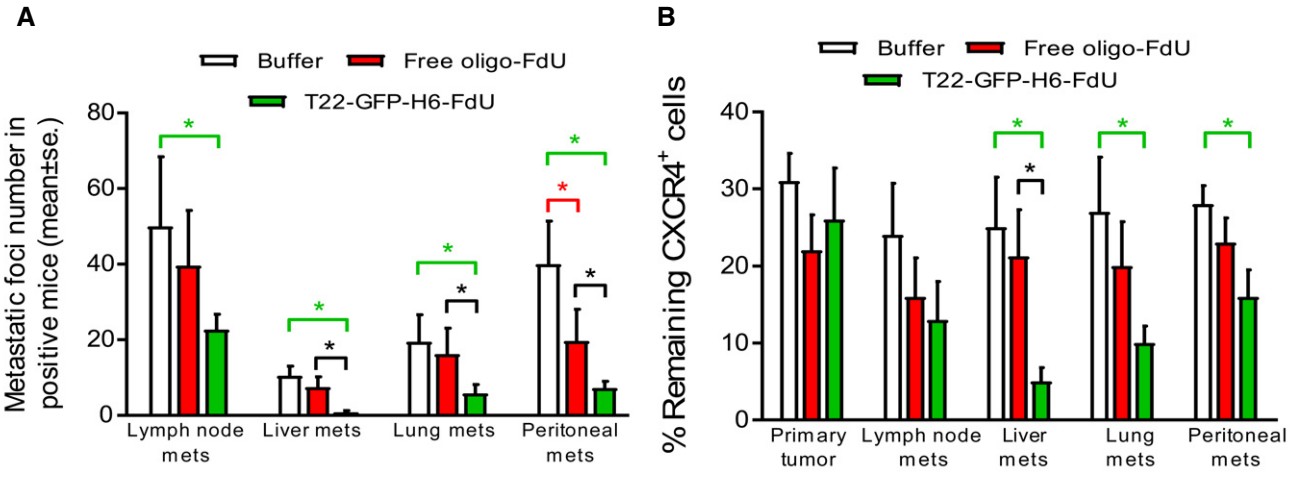

**M5 patient-derived model**

**Remaining CXCR4⁺ Cancer Cell Fraction (CXCR4⁺ CCF)**

**Figure 6.**

**Figure 6.   T22-GFP-H6-FdU prevents metastasis in the M5 patient-derived model in a CXCR4-dependent manner.**

A    T22-GFP-H6-FdU prevents metastases in the CXCR4$^+$ patient-derived M5 model by potently reducing the total and mean number of liver, lung, and peritoneal Mets, as recorded in H&E-stained histology sections at the end of treatment, in comparison with free oligo-FdU or Buffer treatment. In contrast, the number of LN Mets is not reduced after T22-GFP-H6-FdU or free oligo-FdU administration (N = 6 mice per Buffer group; N = 7 mice per free oligo-FdU group; and N = 8 mice per T22-GFP-H6-FdU group; 3 samples/mouse). Data expressed as mean ± s.e.m. Comparison of metastatic foci number by site between groups: (B: Buffer; F: free oligo-FdU; T-F: T22-GFP-H6-FdU). *P*-values for statistical differences: T-F vs. B: *P = 0.006 for LN Mets; *P = 0.001 for LV Mets; *P = 0.003 for LG Mets; *P = 0.001 for PTN Mets (green lines), F vs. B: *P = 0.015 for PTN Mets (red line), T-F vs. F: *P = 0.001 for LV Mets, *P = 0.022 for PTN Mets (black line). Mann–Whitney U-test. See Table 1 for the recording of the percent of metastasis-free mice (mice with undetectable metastases at the end of treatment, and therefore with an absence of CXCR4$^+$ tumor cells) after T22-GFP-H6-FdU treatment. Also, Table 1 describes the reduction in mean foci number and foci size in metastasis-positive mice after T22-GFP-H6-FdU treatment, as compared to Buffer or free oligo-FdU.

B    T22-GFP-H6-FdU induces a higher reduction in CXCR4$^+$ cancer cell fraction (CXCR4$^+$ CCF) in liver, lung, and peritoneal metastatic tissue, at the end of treatment, than free oligo-FdU, as measured by anti-CXCR4 IHC. In contrast, T22-GFP-H6-FdU or free oligo-FdU does not reduce the CXCR4$^+$ CCF in LN Mets or primary tumor tissue after therapy (N = 6 mice per Buffer group; N = 7 mice per free oligo-FdU group; and N = 8 mice per T22-GFP-H6-FdU group; 3 samples/mouse). Data expressed as mean ± s.e.m. Comparison of remaining CXCR4$^+$ CCF by site between groups: (B: Buffer; F: free oligo-FdU; T-F: T22-GFP-H6-FdU). *P*-values for statistical differences: T-F vs. B: *P = 0.012 for LV Mets, *P = 0.027 for LG Mets; *P = 0.038 for PTN Mets (green lines), T-F vs. F: *P = 0.013 for LV Mets (black line). Mann–Whitney U-test.

C    Representative CXCR4 IHC images of the reduction in CXCR4$^+$ CCF induced by T22-GFP-H6-FdU (or its absence in free oligo-FdU mice) at the end of treatment, in the M5 patient-derived CRC model, which quantitation is reported in panel (B). In the M5 model, the highest reduction in foci number and size occurs in liver metastases, which show the highest reduction in CXCR4$^+$ CCF. Note the correlation between the reduction in CXCR4$^+$ CCF induced by T22-GFP-H6-FdU and its antimetastatic effect at each site, measured as number of liver, lung, or peritoneal Mets (Table 1) in the M5 metastatic CRC models [as it happens in the SW1417 model (Appendix Fig S8)]. Note in both Table 1 and Appendix Table S1 that 83% of mice in the T22-GFP-H6-FdU group remained free of liver, lung, or peritoneal metastases at the end of treatment in the SW1417 CRC model, whereas in the M5 CRC model these parameters were in the 38–63% range. Scale bar, 100 μm. Asterisks, tumor tissue; N, normal tissue; LN, lymphatic metastasis.

Mets. Thus, the repeated dose administration at the CXCR4 expression peak in tumors (q3d) achieved our goal of maintaining the CXCR4$^+$ CCF absent, or low, leading to a complete elimination of Mets (achieving, therefore, a complete elimination of CXCR4$^+$ cancer cells) in a portion of mice, in both cell line and patient-derived CCR models. Based on our previous findings of high T22-GFP-H6 nanoparticle internalization in high CXCR4-expressing PTN Mets and low internalization in low CXCR4-expressing LN Mets in CRC mouse models (Céspedes *et al*, 2016), our results suggest that the high T22-GFP-H6-FdU antimetastatic effect observed in peritoneal metastases may come from its high internalization in these foci, leading to high intracellular FdU concentration and DNA damage above DNA repair capacity, triggering in turn high cell killing and a reduction in foci number as well as in CXCR4$^+$ CCF at the end of treatment. This is opposite to the observation of a low internalization of the nanoconjugate and lack of antimetastatic effect and no reduction in CXCR4$^+$ CCF in primary tumor or LN Mets. These findings are reminiscent of the association between antitumor effect by inhibition of Bmi-1 self-renewal protein and reduction in the Bmi1$^+$ CSCs fraction (Kreso *et al*, 2014), and identify the

CXCR4$^+$ CCF in cancer tissue as a possible marker for monitoring metastasis response to T22-GFP-H6-FdU therapy.

In sharp contrast, an equimolecular dosage of unconjugated FdU did not prevent metastases nor reduced the CXCR4$^+$ CCF in metastatic tissue, which may relate to its unselective biodistribution, reaching low FdU concentration in MetSCs and triggering an insufficient level of DNA damage or apoptosis. Despite T22-GFP-H6-FdU also induced a higher level of regression of established LG Mets than free oligo-FdU and a similar effect on LN Mets, our results suggest that T22-GFP-H6-FdU may be more effective at blocking metastasis early, while disseminating from the primary tumor or during secondary organ colonization, than at inhibiting metastatic growth. The significant reduction in tumor emboli intravasation in peri-tumoral vessels observed in T22-GFP-H6-FdU-treated tumors supports this argument. This is also consistent with the functions described for CXCR4 during early metastatic dissemination, including tumor cell trafficking at the invasion front, intravasation, extravasation, or organ colonization in CRC, as described in other tumor types (Zeelenberg *et al*, 2003; Gassmann *et al*, 2009; Hernandez *et al*, 2011; Jin *et al*, 2012a,b; Wendel *et al*,

**Figure 7.   Negligible T22-GFP-H6-FdU biodistribution or toxicity on non-tumor tissues.**

A    Undetectable T22-GFP-H6-FdU emitted fluorescence in normal tissues, except for a transient accumulation 5 h after a 100 μg dose in the liver, which disappears at 24 h. Liver emitted fluorescence is transient and significantly lower than the one registered in tumor tissue. Tumor/Liver ratio = 7.5 (see tumor intensity in Fig 2B, which was registered in the same experiment; N = 5 mice/group). Scale bar, 1 cm. Color key, radiant efficiency units.

B    Representative images depicting the level of DNA double-strand break (DSB) induction in histologically normal bone marrow 5 h after treatment, as measured by anti-γ-H2AX, which is higher in free oligo-FdU-treated mice than in T22-GFP-H6-FdU (P = 0.047). Low level of cells containing DSBs in histologically normal kidney after T22-GFP-H6-FdU or free oligo-FdU treatment, a finding occurring in all normal tissues analyzed (N = 50, 5 mice/group; 10 fields/mouse). Scale bar, 100 μm.

C    Representative images showing lack of histopathological alterations in H&E-stained tissue or apoptotic induction in H&E-stained samples of CXCR4$^+$ (bone marrow) and CXCR4$^-$ (brain, kidney, liver, lung, and heart) normal tissues 24 h after the administration of a 100 μg dose of T22-GFP-H6-FdU or an equimolecular dose of free oligo-FdU (N = 5/group). Note that the transient nanoconjugate distribution to liver or the DNA damage induced in bone marrow does not lead to cytotoxicity on these non-tumor tissues (N = 50, 5 mice/group; 10 fields/mouse). Scale bar, 100 μm.

D    Lack of differences in body weight among groups registered along time in the SW1417-derived CCR model and the regression of metastases protocol (mean ± s.e.m., N = 10 mice/group).

E    Lack of differences in body mouse weight among groups registered along time in the SW1417 cell line-derived model and the prevention of metastasis protocol [mean ± s.e.m., Buffer (N = 11; free oligo-FdU (N = 12), T22-GFP-H6-FdU (N = 12)].

F    Lack of differences in body mouse weight among groups registered along time in the M5 patient-derived model and the prevention of metastasis protocol [mean ± s.e.m., Buffer (N = 6); free oligo-FdU (N = 17); T22-GFP-H6-FdU (N = 8)].

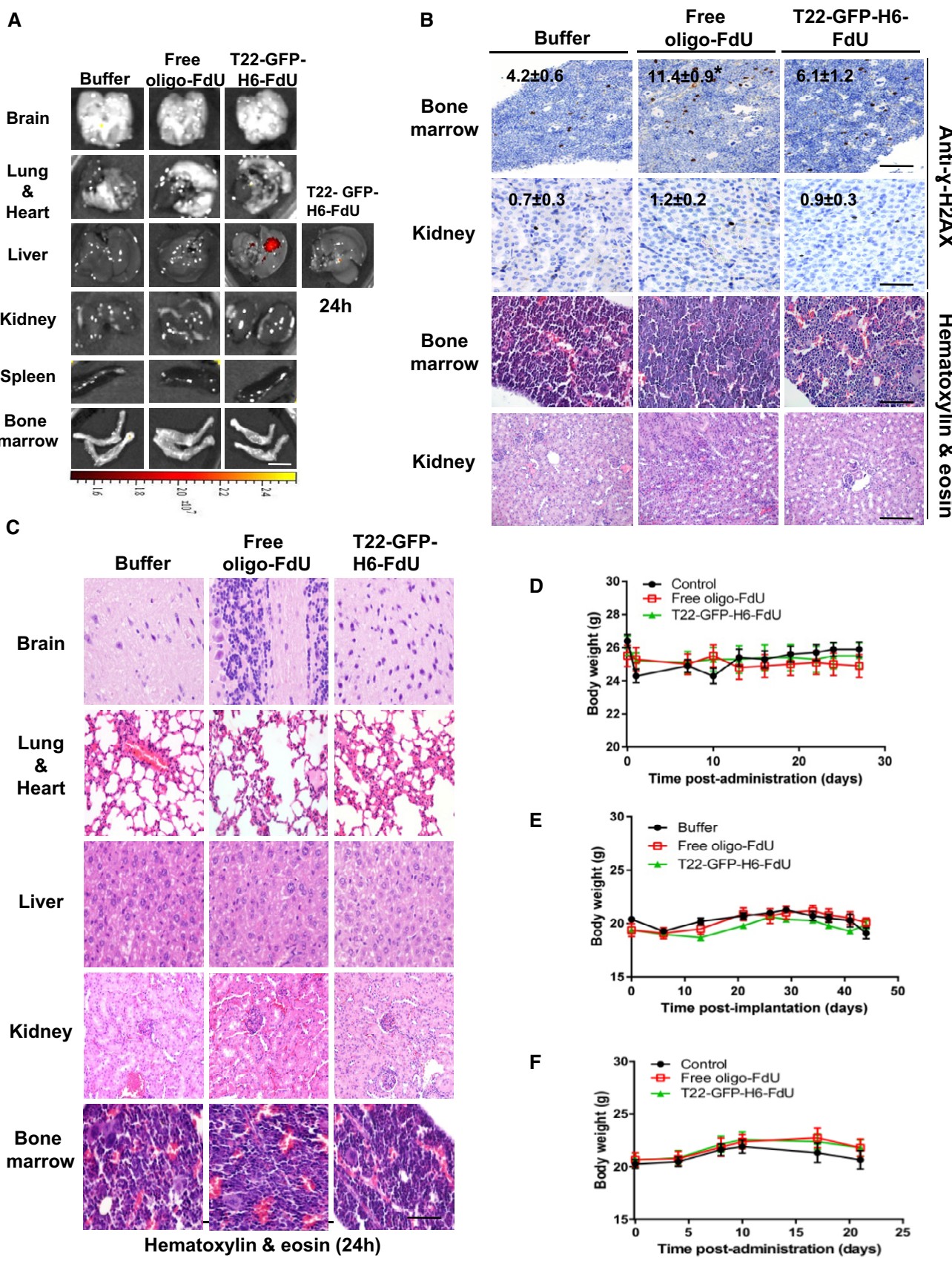

**Figure 7.**

2012). As expected, this nanoconjugate shows also low potency at controlling primary tumor growth, which relates to self-renewal rather than to cell trafficking (Steeg, 2016; Brabletz et al, 2005; Sleeman & Steeg, 2010; Oskarsson et al, 2014). This is also in agreement with CRC CD133$^+$CXCR4$^+$ CRC cells displaying CSC and MetSC capacities, whereas CD133$^+$ CXCR4$^-$ cells have only CSC capacity (Zhang et al, 2012), which further supports the notion that T22-GFP-H6-FdU selectively targets the subset of CXCR4$^+$ MetSCs rather than CSCs.

Our nanoconjugate displayed also a high therapeutic window, since we achieved selective CXCR4$^+$ tumor cell uptake and high antimetastatic effect while achieving negligible distribution (as reported for T22-GFP-H6; Céspedes et al, 2016) or histological alterations in normal tissues (expressing or not CXCR4) with no sign of toxicity or body weight lost. This is consistent with the very high level of CXCR4 expression in poor prognosis CRC (Kim et al, 2005, 2006; Schimanski et al, 2005) and other neoplasias (Fischer et al, 2008; Nimmagadda et al, 2009; van den Berg et al, 2011) in comparison with non-tumor tissues.

In summary, we demonstrated that targeted drug delivery to metastatic stem cells (MetSCs) aimed at their selective depletion is a clinically relevant and reachable therapeutic goal for metastasis control in CRC, with no associated toxicity, when using protein-based nanoconjugates such as T22-GFP-H6-FdU. To our knowledge, this is the first compound that achieves selective antimetastatic effect, which supports a change in drug development focus from primary tumor to metastasis control (Steeg & Theodorescu, 2008; Weber, 2013; Steeg, 2016) that is expected to highly enhance its clinical impact, considering that metastases continue causing most cancer deaths (Mehlen & Puisieux, 2006; Spano et al, 2012; Riihimäki et al, 2016). Our results support its use mostly in the neoadjuvant setting to achieve early-stage metastasis control in non-metastatic high-risk patients or in patients with limited disease as proposed (Steeg, 2012, 2016; Weber, 2013). We also expect to increase therapeutic precision in the use of the nanoconjugate by selecting candidate patients with high CXCR4 overexpression, who are likely to respond, and determining CXCR4$^+$ CCF along time to monitor their response to treatment. A similar therapeutic approach could be implemented for metastasis control in a variety of cancer types in which CXCR4$^+$ MetSCs associate with poor prognosis (Balkwill, 2004; Kucia et al, 2005; Hermann et al, 2007).

# Materials and Methods

### Synthesis of the T22-GFP-H6-FdU therapeutic nanoconjugate

T22-GFP-H6 is a protein nanoparticle produced in bacteria using a recombinant DNA strategy, as previously described (Unzueta et al, 2012a). The nanoconjugate was synthesized by covalent binding of the targeting vector and oligo-FdU, a pentameric oligonucleotide of Floxuridine (5-Fluoro-2′-deoxyuridine; Sigma-Aldrich Chemie GmbH, Steinheim, Germany), both functionalized before their conjugation. The oligo-FdU was functionalized with a thiol group as described in Appendix Fig S1, and T22-GFP-H6 was functionalized by reacting with the linker 4-maleimido hexanoic acid N-hydroxy-succinimide ester (Thermo Fisher, Waltham, MA, USA), following

the protocol for biofunctionalization of proteins described by Hermanson (2013). This linker binds the amino groups of the external lysines of the T22-GFP-H6 protein adding maleimido groups. The final T22-GFP-H6-FdU nanoconjugate was obtained reacting T22-GFP-H6 functionalized with maleimide and oligo-FdU-thiol (Michael reaction; Nair et al, 2014). The final reaction product was purified by dialysis, as previously described for T22-GFP-H6 (Unzueta et al, 2012b). The functionalization and physico-chemical characterization of oligo-FdU with thiol are described in Appendix Fig S1. The physico-chemical and functional characterization of the reaction products for the synthesis of T22-GFP-H6-FdU is described in Appendix Fig S2.

### CXCR4 and SDF-1α expression in SW1417 cells

CXCR4$^+$ luciferase$^+$ SW1417 CRC cells expressing the luciferase reporter gene (derived from the parental SW1417 human colorectal cell line, ATCC® CCL238™, Manassas, VA, USA) were cultured in modified Eagle's medium (Gibco, Thermo Fisher, Waltham, MA, USA) supplemented with 10% fetal calf serum (Gibco) and incubated at 37°C and 5% $CO_2$ in a humidified atmosphere.

Fluorescence-activated cell sorting (FACS) analysis was performed in duplicate to verify cell surface expression of CXCR4. Briefly, one million CXCR4$^+$ luciferase$^+$ SW1417 cells were washed in phosphate-buffered saline containing 0.5% bovine serum albumin (PBS–BSA) and incubated for 30 min at 4°C with PE-Cy5 mouse anti-human CXCR4 monoclonal antibody or PE-Cy5 mouse IgG2a as an isotype control (BD Biosciences, Becton Dickinson, Franklin Lakes, NY, USA). Unbound antibody was removed by two washes with PBS–BSA. Data acquisition was performed using flow cytometry (FACSCalibur, Becton Dickinson, Franklin Lakes, NY, USA) and analyzed by Cell Quest Pro software.

To quantify SDF-1α release, $5 \times 10^4$ CXCR4$^+$ luciferase$^+$ SW1417 or 1BR3.G fibroblasts (SDF-1α-expressing cells used as control; ECACC, Cat. No. 90020507, Salisbury, UK cells) were cultured in DMEM with 10% FBS on a 24-well plate, for 48 or 72 h. Media was recovered at these times to measure their level of SDF-1α using a commercially SDF-1α ELISA kit (RayBiotech, Norcross, GA, USA). Experiments were performed in duplicate.

### T22-GFP-H6-FdU internalization, CXCR4 specificity, and cytotoxicity in CXCR4$^+$ cells *in vitro*

We assessed the internalization capacity of the nanoconjugate by exposing CXCR4$^+$ luciferase$^+$ SW1417 cells for 1 h to 1 μM T22-GFP-H6-FdU concentration, treating them with 1 mg/ml trypsin (Gibco, Waltham, MA, USA) for 15 min, and measuring the green emitted fluorescence of the internalized nanoconjugate particles in the FACSCanto system cytometer (Becton Dickinson, Franklin Lakes, NJ, USA), using a 15 mW air-cooled argon ion laser at 488 nm excitation. Fluorescence emission was measured with a D detector (530/30-nm band-pass filter).

To assess specificity for CXCR4 receptor-mediated internalization, we performed competition studies incubating CRC SW1417 cells with the CXCR4 antagonist AMD3100 (Sigma-Aldrich, Saint Louis, MO, USA) in a 1:10 (protein:antagonist) molar ratio for 1 h before exposure to the nanoconjugate at 1 μM for an additional hour.

T22-GFP-H6-FdU subcellular localization was performed by culturing the cells in MatTek culture dishes (MatTek Co., Ashland, MA, USA); then, T22-GFP-H6-FdU was added in OptiPro medium supplemented with L-glutamine. The nuclei were labeled with 0.2 μg/ml Hoechst 33342 (Molecular Probes, Eugene, OR, USA) and the plasma membranes with 2.5 μg/ml CellMaskTM Deep Red (Molecular Probes, Eugene, OR, USA) for 10 min in the dark and washed in phosphate-buffered saline (Sigma-Aldrich). Live cells were recorded by TCS-SP5 confocal laser scanning microscopy (Leica Microsystems, Wetzlar, Germany) using a Plan Apo 63×/1.4 (oil HC × PL APO lambda blue) objective. To determine particle localization inside the cell, stacks of 10–20 sections for every 0.5 μm of cell thickness were collected and three-dimensional models were generated using Imaris version 7.2.1 software (Bitplane, Zurich, Switzerland).

Next, we studied T22-GFP-H6-FdU cytotoxic activity measuring cell viability and using the MTT metabolic test (Roche, Basel, Switzerland), following manufacturer recommendations. To that purpose, we exposed SW1417 CRC cells to T22-GFP-H6-FdU at 1.0–1,000 nM concentration range and measured their viability at 72 h as compared to equimolecular concentrations of T22-GFP-H6 or free oligo-FdU. We then construct a dose–response curve and determine the linearized T22-GFP-H6-FdU dose–response trend line for each compound.

**Generation of CCR mouse models**

Experiments were approved by the Mouse Ethics Committee at Hospital de la Santa Creu i Sant Pau.

We used three different CRC mouse models, one generated by subcutaneous CRC cell implantation to study nanoconjugate biodistribution and induction of CRC apoptosis, and two generated by orthotopic cell implantation to study the antimetastatic effect, either for prevention of metastases or regression of established metastases. To generate two of these models, we used 5-week-old *Swiss nude* mice, whereas in one model we used *NOD-SCID* mice. They were all female mice weighing 18–20 g (Charles River, L'Arbresle, France) and were housed in a sterile environment with bedding, water, and γ-ray-sterilized food *ad libitum.*

*Subcutaneous (SC) mouse CRC model*
We generated a subcutaneous CRC model injecting $1 \times 10^7$ CXCR4[+] luciferase[+] SW1417 human CRC cells (expressing luciferase to allow bioluminescence monitoring of tumor growth) re-suspended in 250 μl of media in the mouse flank. When tumors reached 700 mm³ were excised and implanted SC tumor aliquots ($3 \times 3 \times 3$ mm) by the trocher system in a cohort of mice. SC model was used to assess tumor uptake, nanoconjugate internalization, and *in vivo* competition studies by co-administration of the CXCR4 antagonist AMD3100. It was also used to determine the induction of DNA double-strand breaks, tumor cell apoptosis, and the fraction of CXCR4[+] cancer cells remaining in tumor tissue (CXCR4[+] CCF) along time after treatment, as described below. These data were used to design the required dosage interval for the nanoconjugate repeated dose therapy in subsequent experiments aimed to determine its antimetastatic effect.

*Orthotopic (ORT) CRC mouse model used to study regression of established metastases*
*Swiss nude* mice were anesthetized with ketamine and xylazine, exteriorizing their cecum by a laparotomy. $2 \times 10^6$ CXCR4[+]

luciferase[+] SW1417 CRC cells (expressing luciferase, to allow *ex vivo* bioluminescent identification of metastatic foci in affected organs) were suspended in 50 μl of modified Eagle's medium and loaded into a sterile micropipette. We slowly injected the cell suspension, under a binocular lens, with an approximate 30° angle, and its tip introduced 5 mm into the cecal wall (intracecal microinjection) as described (Céspedes *et al*, 2007). We sealed the entry injection point using BioGlue Surgical Adhesive (CryoLife Inc, Kennesaw, GA, USA) to avoid the reflux of implanted cells and to ensure that no seeding in the peritoneal wall occurred during the procedure. This model was used to evaluate the capacity of the T22-GFP-H6-FdU nanoconjugate to induce the regression of established metastases.

*Orthotopic (ORT) CRC mouse model to study prevention of metastases*
We generated an efficient metastatic model in *NOD/SCID* mice that received an intracecal microinjection (ORT) of CXCR4[+] luciferase[+] SW1417 CRC cells disaggregated from SC tumors previously generated in a different cohort of *NOD/SCID* mice. We also generated a highly efficient metastatic model in NSG mice that received an intracecal microinjection of M5 patient-derived CRC tumor cells disaggregated from SC tumors previously generated in a different cohort of NSG mice (SC + ORT models) as previously described (Alamo *et al*, 2014; Rueda *et al*, 2015). Briefly, when SC tumors reached a volume of 700 mm³, mice were sacrificed by cervical dislocation and tumors were excised, discarding the necrotic areas, and 300 mg of viable tumor tissue was then cut into pieces and disaggregated in a mix of 0.05% trypsin (Invitrogen) and 100 mg/ml DNase (Sigma-Aldrich). The mix was pipetted 30 times, using a 10-ml pipette, and incubated for 10 min at 37°C with shaking. It was then re-pipetted 30 times, using 10-, 3-, and 1- pipettes, and re-incubated for 5 min at 37°C with shaking. The obtained SW1417 single-cell or M5 patient-derived cell suspensions were filtered through a cell strainer and centrifuged at 1,000 *g* for 10 min before counting the cells. We then microinjected $2 \times 10^6$ cells, previously grown in culture and re-suspended in 50 μl of media, in the cecum of each mouse. These models were used to evaluate the capacity of the T22-GFP-H6-FdU nanoconjugate to prevent metastasis development.

**Evaluation of Spheroid formation capacity**

To evaluate spheroid formation capacity after treatment, mice bearing SW1417 CXCR4[+] luciferase[+] or M5 patient-derived tumors were treated with 100 μg of T22-GFP-H6-FdU, the equimolar dose of free oligo-Fdu or vehicle for two consecutive days (N = 2 mouse/group). At 24 h post-treatment, tumors were excised and mechanically and enzymatically disaggregated with a mixture of collagenase IV (0.5 mg/ml) and DNAsa (0.1 mg/ml) in DMEM media at 37°C (Alamo *et al*, 2014). Isolated cells were cultured at $10^6$ cells for 48 h in cancer stem cell media consisting in DMEM/F12, N2 supplement 1×, Hepes 1 M, L-glutamine, glucose 45% (Invitrogen, Carlsbad, CA, USA) Trace elements B and C (1/1,000×; VWR, Barcelona, Spain), 2 μg/ml heparine, 10 μg/ml insulin (Sigma-Aldrich), 0.02 μg/ml human EGF, 0.01 μg/ml human, and b-FGF (Peprotech, London, UK) and using T25 ultralow attachment surface coating flasks (Gibco) in order to minimize cell adherence. We measured spheroid formation by counting the number of spheroids generated from

isolated and cultured cells derived from treated tumors under a contrast phase microscope (200× of magnification) ($N = 8$; 2 mice/group; 4 plates/mice).

### Evaluation of tumor re-initiation capacity

To assess tumor formation capacity after treatment, we inoculated 5 million cells per flank (right and left) in NSG mice that were obtained from the disaggregated M5 tumors generated in these mice after being treated with 100 μg T22-GFP-H6-FdU or the equimolar dose of FdU for two consecutive days ($n = 4$ tumors/group). We registered tumor burden at the point injection site every 2 days and tumor volume using a caliper, measuring the two perpendicular diameters [short diameter (S) and large diameter (L)] of each tumor and applying the following formula: Tumor volume = $L*S^2/2$.

### T22-GFP-H6-FdU tumor uptake, tumor cell internalization, and induction of DNA damage and apoptosis in vivo

We used the SC CXCR4$^+$ SW1417 CRC model to assess the internalization of the T22-GFP-H6-FdU nanoconjugate into the cytosol of CXCR4$^+$ tumor cells after the administration of 100 μg T22-GFP-H6-FdU as an i.v. single bolus compared with Buffer, T22-GFP-H6 (untargeted nanoconjugate), and oligo-FdU (unconjugated free drug). Two, 5, and 24 h after the administration, we euthanized the mouse, resected the tumor, and registered ex vivo the intensity of the green fluorescence emitted by the nanoconjugate that had biodistributed to tumor tissue, using the IVIS® 200-Spectrum (PerkinElmer, Waltham, MA, USA). Following, we took tumor tissue samples and performed immunofluorescence (IF) and immunohistochemistry (IHC) to assess the presence or absence of the corresponding nanoconjugate in the membrane and/or cytosol of tumor cells using an anti-GFP antibody (1:300; Santa-Cruz Biotechnology, CA, USA). We also assess nanoconjugate CXCR4-dependence for internalization, as previously described for the T22-GFP-H6 nanoparticle in (Unzueta et al, 2012a).

The presence and localization of T22-GFP-H6-FdU nanoconjugate (detecting its GFP domain) were assessed by immunofluorescence labeling of formalin-fixed paraffin-embedded (FFPE) samples using standard protocols. Primary antibodies anti-CXCR4 (1:300, Abcam, Cambridge, UK) and anti-GFP (1:250, Abcam, UK) were incubated ON at 4°C. Then, we used the secondary antibodies: chicken IgG-Cy2 for GFP and rabbit IgG-Cy3 for CXCR4. Slides were then stained with DAPI (1:10,000 in TBS) for 10 min RT, rinsed with water, mounted, and analyzed under fluorescence microscope (405 nm, 488 Cy2 and 532/561 filters). Representative pictures were taken using confocal Leica TCS SPE at 200× or 600× magnifications. Immunofluorescence measurements were performed using the ImageJ software. Data were expressed as mean area ± s.e.m (μm$^2$) ($N = 5$ mice/group).

Once, we determined the specific internalization of T22-GFP-H6-FdU in CXCR4$^+$ cancer cells in tumor tissue, we assessed if this nanoconjugate also induced genotoxic damage and apoptosis in tumor tissue, before studying if both activities lead to CXCR4$^+$ cancer cell elimination. To that purpose, we treated the SC SW1417 mouse model, with a single 100 μg iv dose of T22-GFP-H6-FdU, or equimolar doses of T22-GFP-H6, free oligo-FdU, or Buffer. Five hours later, we sacrificed the mice and counted the number of cells containing double-strand breaks (DSBs) in tumor tissue, as assessed

by IHC using an anti-γ-H2AX mAb (1:400, Novus Biologicals, Cambridge, UK) as previously described (Kuo & Yang, 2008; Geng et al, 2010; Podhorecka et al, 2010). To that purpose, we counted the number of cells containing nuclei that stained positive for DSBs, using anti-γ-H2AX mAb IHC, in ten 400× magnification fields in one tumor section per mouse ($N = 5$ mice/group).

We also compared the capacity of T22-GFP-H6-FdU for apoptosis induction after the administration of an equimolecular dose of, T22-GF-H6, free oligo-FdU, or Buffer. Apoptotic signaling was assessed 5 h after treatment, by evaluation of cells positive for active cleaved caspase-3 as measured by IHC, whereas apoptotic induction was assessed 24 h after treatment, by counting the number of condensed or defragmented nuclei, after Hoechst staining. Both parameters were measured in ten 400× high-power magnification fields, in different sections from each tumor, using the Olympus DP73 digital camera.

### Definition of the optimal dose interval by changes in CXCR4$^+$ tumor cell number after T22-GFP-H6-FdU administration

We also used the SC SW1417 mouse model, to determine the capacity of the nanoconjugate to induced DNA damage and apoptosis in tumor tissue, and its relationship with the kinetics of CXCR4 expression levels in the membrane of tumor cells after treatment, regarding the fraction of CXCR4 expressing tumor cells and their intensity, since CXCR4$^+$ are the target cells for the nanoconjugate. To that purpose 24, 48, and 72 h after the administration of a single i.v. bolus of 100 μg T22-GFP-H6-FdU, we euthanize the mice, took tumor samples, fix, and paraffin-embedded them to determine the levels and the percent of tumor cells expressing CXCR4 using IHC with an anti-CXCR4 antibody (1:300, Abcam, UK) as previously described (Céspedes et al, 2014). We used mice treated with equimolecular dosages of T22-GFP-H6, free oligo-FdU, or Buffer, in which we also determined the levels of CXCR4 expression in tumor tissue at the different times points. The results of the kinetics of CXCR4 expression in tumor cells were used to establish the optimal T22-GFP-H6-FdU nanoconjugate administration interval in the repeated dose schedule used to evaluate antimetastatic effect.

### Evaluation of tumor emboli intravasation capacity in early metastatic experiment

We used the M5 patient-derived CRC model in NSG mice to evaluate the capacity of the nanoconjugate to block tumor emboli intravasation. NSG mice received an intracecal microinjection of M5 patient-derived CRC tumor cells ($2 \times 10^6$) disaggregated from SC tumors as described in the above section. At day 7 post-inoculation, mice were treated with i.v. bolus of 100 μg T22-GFP-H6-FdU or the equimolar dose of free oligo-FdU or Buffer for two consecutive days and euthanized 24 h later ($N = 5$/group). The primary tumor was collected, fix, and paraffin-embedded for histopathology evaluation. Intravasated tumor emboli were determined by microscopic analysis under 200× magnification in H&E-stained primary tumor sections. We counted the vascular (blood or lymphatic) invasion by identifying tumor emboli that were invading an endothelium-lined vessel-like structure within the submucosa of the colonic wall, where a majority of vasa are located, and recording their number. Identification of

tumor emboli was done in H&E-stained tumor sections, under 200× magnification in each section, in five different mice per group. CXCR4 expression per tumor emboli was evaluated by using anti-CXCR4 IHC and calculating H-score (multiplying percent of CXCR4$^+$ cells out of total cell number in the emboli area by their staining intensity), scoring each from 0 to 3 (where three is the maximal intensity) per tumor emboli area ($N$ = 5 mice/group) using Cell^D Olympus software (v3.3.).

### Treatment protocol for the evaluation of T22-GFP-H6-FdU induction of metastasis regression

We used the orthotopic and metastatic CRC model developed in *Swiss nude* mice to perform experiment of metastasis regression. We randomized 40 mice in a non-blinded manner into four groups: Buffer, T22-GFP-H6, T22-GFP-H6-FdU, and free oligo-FdU ($n$ = 10/group) and administered repeated i.v. boluses at equimolecular doses, as follows: T22-GFP-H6-FdU: 20 μg, free oligo-FdU: 2.6 nmols, or Buffer, every 3 days (q3d) for a total of 10 doses. We initiated the T22-GFP-H6-FdU administration 2 months after tumor cell implantation, the time at which we determined, in previous experiments, that lymph node and lung metastases were present (see Appendix Fig S4). The experiment was finished when the first animal of the Buffer-treated group required to be euthanized. See below the studied parameters to evaluate the antimetastatic effect.

### Treatment protocol for the evaluation of T22-GFP-H6-FdU metastasis prevention effect

We used the SC + ORT metastatic SW1417 CRC model developed in *NOD/SCID* or the M5 patient-derived CRC model in *NSG* mice to evaluate the capacity of the nanoconjugate for metastasis prevention. In each experiment, we non-blinded randomized mice into three groups: Buffer ($n$ = 7–11), T22-GFP-H6-FdU ($n$ = 8–12), and free oligo-FdU ($n$ = 7–8) and administered repeated i.v. boluses at equimolecular doses, as follows: T22-GFP-H6-FdU: 20 μg; free oligo-FdU: 2.6 nmols; or Buffer), every 3 days (q3d) for a total of 12 doses in the SW1417 model or seven doses in M5 model. We initiated the T22-GFP-H6-FdU administration 1 week after tumor cell implantation before metastatic dissemination has occurred (see Appendix Fig S4). The experiment was finished when the first animal of the Buffer-treated-group required to be euthanized.

### Evaluation of antimetastatic effect and determination of the CXCR4$^+$ cancer cell fraction in tumor tissue at the end of treatment

At the end of both, the regression and the prevention of metastasis, experiments, we applied the same methodology to determine T22-GFP-H6-FdU antimetastatic effect. At necropsy, we recorded the number and size of visible metastasis in the organs where dissemination is expected in colorectal cancer (lymph nodes, liver, lung, and peritoneum) for each mouse in all compared groups. In the luciferase$^+$ SW1417-derived CRC model, we also counted *ex vivo* the number of metastatic foci that emitted bioluminescence in the target organs for metastasis, using the IVIS® 200-Spectrum (PerkinElmer).

We collected and processed samples for histopathological and immunohistochemical analyses. Two independent observers analyzed H&E-stained samples to count the number and measure the size of all observed metastatic foci in sections of each organ in each mouse. We used an Olympus microscope with the Cell^D Olympus software (v3.3) to take images and perform the measurements.

We determine CXCR4 expression in tumor tissue, using IHC with an anti-CXCR4 antibody, as described above, to determine the fraction of CXCR4$^+$ cancer cells remaining in tumor tissue (CXCR4$^+$ CCF) after treatment, including primary tumor and metastatic foci at the different organs affected by metastases (peritoneum, liver, lung, and lymph nodes). The obtained results were used to study a possible correlation between CXCR4$^+$ CCF and antimetastatic effect at the different sites.

### T22-GFP-H6-FdU biodistribution and toxicity in normal organs

We assessed T22-GFP-H6-FdU uptake measuring the green fluorescence emitted by the GFP domain of the nanoconjugate, as well as DNA DSBs and apoptotic induction in normal (non-tumor) tissues using the methodology described above. In addition, two independent observers evaluated the possible histopathological alterations observed in H&E-stained non-tumor tissue samples, searching for signs of toxicity. These tissues included CXCR4-expressing organs (despite expressing this receptor to a significantly lower level than in tumor tissue) where the nanoconjugate could accumulate such as the bone marrow and spleen and we also evaluated the toxicity in non-CXCR4 expressing organs, especially those in which the unconjugated oligo-FdU such as the liver.

### Statistical analysis

Sample size was defined on the basis of previous preliminary experiments. Neither animals nor samples were excluded from the analyses. Randomization of animals into control and experimental groups was performed using the SPSS program. Histology and immunohistochemical samples were coded so that the researcher that analyzed them did not know to which group they belong to. Normal distribution of the data was tested using the Shapiro–Wilk test. The homogeneity of the variance between groups was tested using the Levene's test. We used the Fisher's exact test to analyze possible differences between control and experimental groups of affected mice regarding metastatic rates at the different organs. The non-parametric tests, Kruskal–Wallis, and *post hoc* pairwise Mann–Whitney $U$ two-sided tests were used to compare number and size of metastatic foci in the affected organs among groups. All quantitative values were expressed as mean ± s.e.m. All statistical tests were performed using SPSS version 11.0 (IBM, New York, NY, USA). Differences among groups were considered significant at a $P$ < 0.05.

**Expanded View** for this article is available online.

### Acknowledgements
This work was supported by Plan Estatal de I+D+I 2013-2016, Instituto de Salud Carlos III and MINECO (co-funding from FEDER, Integrated Project of Excellence PIE15/00028, PI18/00650, PI15/00378, PI15/00272, PI12/00327, PI17/

## The paper explained

### Problem
In colorectal cancer (CRC), adjuvant therapy controls tumor progression and prolongs survival at the expense of high toxicity; however, metastases remain the primary cause of death. There is an urgent need to develop less toxic and more effective antimetastatic agents. CXCR4 receptor-overexpressing (CXCR4[+]) cells are metastatic stem cells (MetSCs) since they initiate metastases. Moreover, high CXCR4 tumor expression associates with metastatic dissemination and confers poor patient prognosis, a finding similar to other cancers. We hypothesized that a protein-based nanoconjugate (T22-GFP-H6-FdU) that selectively delivers the genotoxic drug Floxuridine (FdU) to CXCR4[+] MetSCs will be antimetastatic.

### Results
In contrast to free oligo-FdU, intravenous administration of T22-GFP-H6-FdU selectively accumulates and internalizes in CXCR4[+] cancer cells, triggering DNA damage and apoptosis, which leads to the selective depletion of CXCR4[+] MetSCs and to reduced tumor formation and spheroid formation. As compared to free oligo-FdU, repeated T22-GFP-H6-FdU administration, in cell line and patient-derived CRC mouse models, blocks CXCR4[+] tumor emboli intravasation in colonic peritumoral vessels and completely prevents metastases development in a high percent of mice. In addition, this nanoconjugate induces CXCR4 expression-dependent and site-dependent reduction in foci number and size in liver, peritoneal, or lung metastases in the rest of mice. T22-GFP-H6-FdU induces also higher regression of established metastases than free oligo-FdU, with negligible biodistribution or toxicity in normal tissues, showing, therefore, a high therapeutic index.

### Impact
The observation of a potent antimetastatic effect validates metastatic stem cells (MetSCs) as targets for clinical therapy. Moreover, to our knowledge, this protein-based nanoconjugate is the first compound that achieves a selective antimetastatic effect, which supports a change in drug development focus from primary tumor to metastasis control. This new approach is expected to highly enhance its clinical impact in CRC and other cancer types in which CXCR4 mediates metastasis development. Our results support the use of this nanomedicine in the neoadjuvant setting to achieve early-stage metastasis control in non-metastatic high-risk patients or in patients with limited disease as proposed. Its use could also increase therapeutic precision by selecting candidate patients with high CXCR4 overexpression, who are likely to be sensitive to the nanoconjugate, and determining CXCR4[+] cancer cell fraction along time to monitor their response.

00150, BIO2013-41019-P CTQ2014-52588-R, and BFU2010-17450), AGAUR (2017 SGR 865, 2017 SGR 229, 2014PROD-00055), MaratóTV3 (416/C/2013), La Caixa Foundation, and CIBER-BBN Nanomets and Nanoprother Intramural Projects, and used CIBER-BBN Nanotoxicology and Protein Production Platforms for its development. GOA and NBT groups are part of the Spanish ICTS-141007 NANBIOSIS Network for Nanomedicine. M.V.C and U.U. are supported by Miguel Servet and Sara Borrell contracts from ISCIII, respectively. P.A. obtained a Fellowship from the Josep Carreras Research Institute. A.V. received an ICREA ACADEMIA Award.

## Author contributions
Conception and design: RM, AV, EV, MVC, and UU. Development of methodology: MVC, UU, AA, RE, AS-C, PA, RS, AG, IC, and MAM. Acquisition of data: MVC, UU, AA, EV, RS, PA, AG, and AS-C. Analysis and interpretation of data: RM, AV, RE, EV, MVC, UU, AA, IC, and AL-P. Writing of the manuscript: RM, AV, EV, RE, and MVC. Review and/or revision of the manuscript: UU, RE, IC, MAM, AS-C, PA, AG, RS, AA, and AL-P. Study supervision: RM, AV, and EV.

## Conflict of interest
EV, UU, AV, MVC, IC, and RM are co-inventors of a patent (WO2012095527) covering the use of T22 as an intracellular targeting agent.

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
