## [Review Process File · EMBO Molecular Medicine]

Selective depletion of metastatic stem cells as therapy for human colorectal cancer

María Virtudes Céspedes, Ugutz Unzueta, Anna Aviñó, Alberto Gallardo, Patricia Álamo, Rita Sala, Alejandro Sánchez-Chardi, Isolda Casanova, María Antònia Manges, Antonio Lopez-Pousa, Ramón Eritja, Antonio Villaverde, Esther Vázquez, Ramón Manges.

Review timeline:

Submission date:	13 th December 2017
Editorial Decision:	19 th February 2018
Revision received:	10 th July 2018
Editorial Decision:	25 th July 2018
Revision received:	3 rd August 2018
Accepted:	14 th August 2018

Editor: Celine Carret

Transaction Report:

1st Editorial Decision

19th February 2018

Thank you for the submission of your manuscript to EMBO Molecular Medicine. We have now heard back from the three referees whom we asked to evaluate your manuscript.

You will see from the set of comments pasted below that the referees found the study of interest and recommend major revision. They all agree however that the data needs to be strengthened for the conclusions to be more robust. Nice suggestions are provided by all referees and we would encourage you to revise the paper accordingly to make it stronger and more fitting to our scope.

REFeree REPORTS.

Referee #1 (Comments on Novelty/Model System for Author):

Data analysis has been performed based on biological replicates. Statistical tests proof the significance of the presented effects.

The novelty is medium/high: An AMD3100-mediated delivery of siRNAs to CXCR4 liver cancer has been described previously by Liu et al. in *Molecular Therapy* (2015 Nov; 23(11): 1772-1782). However, a specific effect of a CXCR4 surface receptor-targeted drug on liver and lung metastatic colorectal cancer tumor cells has not been described before to my knowledge.

The medical impact is high since the characterized drug shows little to no side-effects on normal tissue cells. Hence, the drug might be a strong candidate for clinical trials.

The model system is adequate. CRC cell line and primary CRC cells have been used. Importantly, orthotopic transplantation has been performed in order to study the metastatic process of cancer progression. This method has become the state-of-the-art, and it outperforms intra tail-vein or intrasplenic cancer cell injections since it recapitulates the complete metastatic process.

Referee #1 (Remarks for Author):

In their manuscript, María Virtudes Céspedes and colleagues use a nanoparticle-coupled drug (Floxuridine) targeted to the CXCR4 receptor via its T22 ligand. By performing in vitro and in vivo experiments, they show that the nanoparticle-coupled drug enters the tumor cell in a CXCR4-dependent fashion, accumulates in the cytoplasm, and elicits a stronger cytotoxicity than the particle-free drug. Importantly, treatment of tumor xeno-engrafted mice with T22-GFP-H6-FdU effectively reduced CXCR4+ metastatic tumor foci in a 48 hrs time window, while normal tissue cells were not affected by the drug. Interestingly, CXCR4+ cells recurred 72 hrs post-treatment, presumably due to the drug pharmacokinetic, but a repetitive treatment schedule kept the metastatic foci (liver and lung) re-growth at bay. Interestingly, lymph node metastasis did not benefit from T22-GFP-H6-FdU when compared to free Oligo-FdU.

The study provides strong data that support the concept of targeted drug delivery to aggressive CXCR4+ cancer cells. In a pre-clinical model of tumor metastatic outgrowth, the laboratory of Ramón Mangués provides very appealing data showing that T22-GFP-H6-FdU specifically elicits DNA damage and apoptosis in CXCR4+ cancer cells. Normal tissue is spared from this drug which fails to enter the cell in the absence of high-level cell-surface standing CXCR4 receptor. Overall, the study provides a novel anti-metastatic treatment strategy which very likely can be translated to the clinic.

The following points should be addressed to further improve the quality of the manuscript:

Major points

1) T22-GFP-H6-FdU shows a significant effect on lung and liver metastasis formation. However, growth of the primary tumor and lymph node metastasis is largely unaffected. The authors explain this effect by different levels of cancer cell CXCR4 expression in different locations. However, according to the IHC staining provided in Figure 4D, primary tumors and LN metastasis show indeed a reasonable CXCR4 staining which gets reduced after administration of T22-GFP-H6-FdU. Since the authors also describe efficient delivery and drug-uptake by sub-cutaneously implanted cancer cells, it should be expected that these cells, after experiencing increased DNA damage and caspase-cleavage (Fig. 2D), get largely depleted from the primary tumor. The authors should therefore provide more experimental evidence to support their hypothesis regarding the differential impact of T22-GFP-H6-FdU on primary and metastatic tumor growth. Especially, the bio-distribution of the drug to the orthotopic transplantation site and LN metastasis could be indeed worse, therefore lowering the effective drug concentration when compared to the liver or lung.

2) Related to point 1, the possibility exists that CXCR4 positive cells, which due to the obvious tumor heterogeneity (CXCR4+ and CXCR4- cells) represent only a fraction of the tumor, are dispensable for primary tumor and LN metastasis growth. Still, these cells might be key to induce and maintain metastatic tumor growth in the lung or liver. A similar phenotype has been recently described by the laboratory of Frederic Sauvage, showing that depletion of LGR5-expressing CRC stem cells prevents the formation and growth of liver metastasis while primary tumor growth at the orthotopic transplantation site remained unaffected (FS e Melo 2017, Nature, 2017). A study by Weidong Wu et al. (Oncotarget, 2016 Dec6) shows that LGR5/CXCR4 double positive CRC cells show the highest capacity of tumor re-formation in serial transplantation experiments. Hence, T22-GFP-H6-FdU administration might indeed kill CRC stem cells which possess highest tumor/metastasis re-formation capacity. To further clarify this question, the authors should analyse as to whether treatment with T22-GFP-H6-FdU reduces overall LGR5 expression in the primary tumor orthotopic implant. It would be also very interesting to address if primary tumor cells of T22-GFP-H6-FdU treated animals possess a lower tumor-re-initiation capacity. This could be easily addressed by assessing the tumor-organoid or spheroid formation capacity of T22-GFP-H6-FdU-treated (CXCR4 low) vs non-treated (CXCR4 high) primary tumor cells.

Referee #2 (Comments on Novelty/Model System for Author):

To my view the manuscript focuses in a very relevant issue and should provide relevant information about cancer therapy, however the work needs to be technically improved.

Referee #2 (Remarks for Author):

This work focuses in a very interesting subject that is the possibility of specifically targeting/killing

metastasis-initiating cells for anti-cancer therapy. Authors use a conjugate compound that includes a region that binds the CXCR4 receptor that relates with metastatic capacity in several models and a cytotoxic agent that induces DNA damage. Ideally, this compound will target metastatic cells leading to improved therapeutic activity of the toxic drug. However, different technical issues together with the lack of details about procedures, quantifications or the number of replicates performed for each experiment strongly weaken the solidity of the results shown.

Comments and suggestions to the main concerns:

1-Cxcr4 and GFP detection in 1C needs to be done in double IF (fluorescent IHC) to show that GFP is specifically internalized in the CXCR4+ population of the tumor. Quantification of different tumor areas and animals is required.

2- In Figure 3A the number of gammaH2A+ cells is similar in the images corresponding to T22-GFP-H6-FdU and free FdU-treated tumors. However, in the text it is said: the number of DSBs foci in tumors was significantly higher than after free oligo-FdU (22.8 {plus minus} 1.4 versus 13.4 {plus minus} 0.7; p=0.02). If they are referring to the percent of tumors cells that are positive, the sentence needs to be reformulated. This is not foci but cells. Also, I would suggest changing this statement to: "the number of cells (per 20X field????) containing DSBs foci in the tumors was slightly but significantly higher than after free oligo-FdU...". Again, the sentence needs to include details on how these numbers were obtained.

3- In 3A and 3B including a bigger area of the tumor (in addition to the detail) will help to interpret the results, since small areas can be deliberately selected to illustrate any conclusion. Moreover, selective elimination of CXCR4+ cells in the different treatments and at the different periods of treatment has to be quantified from various animals in each group (IHC, flow-cytometry...).

4- An important question that should be tested is whether tumors lacking CXCR4 are capable to metastasize (i.e. tumors treated with T22-GFP-H6-FdU for 24-48 hours) and whether CXCR4+ cells are responsible for metastasis in this particular model. Can authors compare the percent of CXCR4+ cells in the bulk of the tumor, the invasive areas and the metastasis at early stages of invasion?

5-In 3B the quality of the images of basal and treated tumors is totally different. Is there any reason for this heterogeneity?

6-In 4B, graphs are randomly labeled and it seems that the effect of the FdU conjugate is only compared with controls but not with the free FdU-treated animals, although it is not indicated. Also, deviations are lacking. In addition, the signal associated to the metastatic component in the different animals (reflecting metastatic load) and treatments need to be shown, not just the number of foci. The same criticism applies to 4C and in this case it is unclear how the percent of remaining CXCR4+ cells in the metastasis have been calculated. Images supporting these results have to be included in the main figure.

7- In 4D, images are very small specially when compared with previous images. This is even worst in the case of 5B where DNA damage cannot be evaluated at all. In addition, 5B is labeled as anti-g-H2AX (5h) what suggest that animals were treated for 5h with this antibody. If 5h is the period of treatment with the therapeutic compounds, this needs to be better indicated.

8- Flow cytometry analysis of DNA damage and apoptosis in the CXCR4+ and CXCR4- populations of tumors exposed to different treatments (i.e. in 3A and 4D) will help to support the main message of the manuscript.

9- It should be specifically mentioned or tested whether CXCR4 is expressed in any particular tissue of the body and the possible impact that the conjugate can exert in this tissue (if any).

In general, results are shown in a very descriptive manner and even when quantifications are included it is impossible to know how they have been obtained. The number of animals used in the different experiments is not mentioned what difficult obtaining definitive conclusion. Examples are found in 2C, 3A, 3B, 5B, but all along the manuscript.

Referee #3 (Comments on Novelty/Model System for Author):

No endogenous expression levels of the target.

Referee #3 (Remarks for Author):

Selective depletion of metastatic stem cells as therapy for human colorectal cancer
 María Virtudes Céspedes et al.

Summary:

One of the major concerns in the field of colorectal cancer therapy is the massive metastatic spread of the tumors. Metastatic stem cells (MetSCs) form a subset of cancer stem cells that facilitates dissemination of cancer cells, their trafficking and eventually the re-growth of tumor cells away from the primary site. Targeting the MetSCs using nanoparticle based drug delivery is one such approach which is currently widely explored in the cancer field. Cancer cell CXCR4 receptor overexpression has been associated with metastatic properties, tumor growth and poor patient prognosis. Using a drug-nanoconjugate T22-GFP-H6-FdU, specifically targeting CXCR4+ cancer cells, Céspedes et al. here describe the method to selectively target these cells and their elimination leading to antimetastatic effects. The authors study the effects and biodistribution of the nanoconjugate in CRC cell line and patient derived model system. The authors demonstrate a potent and site-dependent metastasis prevention using the nanoconjugate targeting CXCR4+ cells.

Major Points:

- The authors make use of the CXCR4 overexpressing CRC cell line model (CXCR4+ SW1417) to study the internalization of the nanoconjugate. However, they do not address this with respect to cells expressing normal levels of CXCR4 receptor. A cell line with normal expression of CXCR4 should be used as a control. Also, the amount of CXCL12, the ligand for CXCR4+ must be evaluated. The CXCR4/CXCL12 axis is associated with various stages of tumor metastasis. The expression levels of CXCR4 and CXCL12 (with qPCR or SDS PAGE) in the CXCR4+ SW1417 cell line should be addressed before and after the internalization of the nanoconjugate.
- A well characterized role of CXCR4 is in activation of MAPK/ERK and PI3K/AKT signaling. The authors show the selective targeting of the CXCR4+ cells in in vitro and in vivo but do not address the physiological effect on the system. Whether the targeted killing of CXCR4+ cells lead to suppression of downstream signaling of CXCR4 to have an effect on tumor growth needs to be evaluated.
- The authors show the selective tumor biodistribution and internalization of the nanoconjugate in the CXCR4+ SW1417 CRC mouse model (Figure 2), however, they do not show the uptake in a non-tumor tissue from the same model. The CXCR4 is expressed in normal cells as well. They do address the lack of T22-GFP-H6-FdU accumulation in normal tissues in Figure 5, however, it's unclear which mouse model was used for Figure 5. The tumor vs normal tissue should be addressed in the same figure for the smooth flow of the paper.
- The authors show the ability of nanoconjugate to induce double strand breaks (DSBs) to show its capacity to release FdU in target cells to reach nucleus and induce DNA damage (Figure 3A). They look at the foci 5 hours after the treatment. However, by that time DNA damage response (DDR) will also be expected to happen. It will be good to see the foci status at an earlier time point. It's difficult to separate DSB from DDR but a time-series can be performed to state whether it is DSBs or DDR. The localization with other repair proteins can also clear between the two.
- Figure 4 was not easy to follow. The authors need to rewrite the description and arrange the figure panels more clearly. They state on page 8, top line, the histological evaluation of LV, LG.....was done for Met+ mice however in Figure 4 we do not see the representative histology sections. Figure 4B shows the quantification of SW1417 and M5 model but panel A only shows the mets representation from SW1417 model. Also, the switch between their mouse models is random. The writing needs to be clearer on which animal model they refer to. Also, the quantification shown in Figure 4C belongs to panel D but the title to the two panels is misleading. Both are M5-patient derived model.

- Drug leakage is a common problem with nanoparticle derived drug targeting. The authors do not address on how much drug is actually delivered to the tumor cell.

Minor Points:

- The fourth line under the heading "T22-GFP-H6-FdU internalization, CXCR4 specificity and cytotoxicity in CXCR4+ cells in vitro" has a typo with SW141-luc cells (should be SW1417-luc cells).
- The authors do not describe the methodology of the H2AX γ IHC but only refer to the articles. However, the articles cited 'Kuo & Yang, 2008; Podhorecka et al, 2010, on page 5 are reviews and give no details of the method in itself. The authors need to add the details of antibody dilution, incubation time to the method section.
- Based on the histology sections from various tissues in different mouse models, the authors present the quantification of the number of foci in Figure S5B. However, there are no histology representative pictures to verify that. A panel of H&E stained section is present in Fig. S4 but it is not clear if they are representative of the same quantification in Fig. S5B. It will be easy to rearrange the sub-figures accordingly.

1st Revision - authors' response

10th July 2018

***** Reviewer's comments *****

Referee#1 (Comments on Novelty/Model System for Author):

Data analysis has been performed based on biological replicates. Statistical tests proof the significance of the presented effects. The novelty is medium/high: An AMD3100-mediated delivery of siRNAs to CXCR4 liver cancer has been described previously by Liu et al. in *Molecular Therapy* (2015 Nov; 23(11): 1772-1782). However, a specific effect of a CXCR4 surface receptor-targeted drug on liver and lung metastatic colorectal cancer tumor cells has not been described before to my knowledge.

The medical impact is high since the characterized drug shows little to no side-effects on normal tissue cells. Hence, the drug might be a strong candidate for clinical trials. The model system is adequate. CRC cell line and primary CRC cells have been used. Importantly, orthotopic transplantation has been performed in order to study the metastatic process of cancer progression. This method has become the state-of-the-art, and it outperforms intra tail-vein or intrasplenic cancer cell injections since it recapitulates the complete metastatic process.

We thank the Reviewer for commenting on the publication by Liu et al, who used CXCR4-targeted lipid-based nanoparticles for VEGF siRNA delivery and treatment of liver cancer. We think, however, that the AMD-NP nanoparticle the authors used did not achieve what the authors claimed. Thus, despite they obtained antimetastatic effect, it was not due to the selective delivery of VEGF.siRNA to CXCR4+ cancer cells. We base this argument on the fact that both, control (scrambled) siRNA-loaded AMD-NPs and VEGF-siRNA-loaded AMD-NPs showed a similarly high reduction in the number of metastatic lung nodules (Fig. 7B of Liu's article), whereas only the VEGF.siRNA-loaded AMD-NPs was capable of downregulating VEGF in tumor tissue. Control-siRNA AMD-NPs did not downregulate VEGF (Fig. 5C of the article) but showed a similarly potent anti-metastatic effect.

In contrast to Liu's AMD-NPs, our protein-based nanoconjugate displays a high targeting capacity, since it achieves high and selective accumulation in CXCR4⁺ tumor cells. Consistently, most of the nanoparticle injected dose (ID) is distributed to tumor (higher than 85% ID), whereas only low or negligible nanoparticle accumulation is observed in normal organs (including the liver) (please, see our previous article by Céspedes et al. 2016). In contrast, lipid-based nanoparticles, such as AMD-NPs, confront barriers that protein nanoparticles do not. One of the main barriers to their selective tumor accumulation is the formation of a "protein corona" after injection in the bloodstream. This corona covers the lipid nanoparticle surface and blocks its targeting capacity (that is, its ability to internalize in target cancer cells expressing a specific receptor (Rosenblum 2018), being instead recognized and accumulated by phagocytic cells in RES organs such as the normal liver (Qi 2017).

This effect may explain why Liu et al. report a higher uptake in normal liver (4% ID) than in tumor tissue (only 2% ID) out of the total injected AMD-NP dose (Fig. 5A). In this regard, it is currently recognized that lipid-based nanoparticles are still far from reaching the clinic (Qi 2017).

In summary, we think that our results are highly novel since we are the first to demonstrate high antimetastatic activity through the highly selective and CXCR4-dependent delivery of the drug Floxuridine to liver, lung and peritoneal metastases, leading to the specific elimination of CXCR4⁺ cancer cells.

We also thank the Reviewer, and agree with him/her, on that the nanoconjugate could be a good candidate for clinical translation. Indeed, these results have pushed us to take steps towards preclinical regulatory testing previous to clinical trials. We are also grateful to the Reviewer's comment on the appropriate use of the orthotopic models to test antimetastatic activity.

Referee #1 (Remarks for Author):

In their manuscript, María Virtudes Céspedes and colleagues use a nanoparticle-coupled drug (Floxuridine) targeted to the CXCR4 receptor via its T22 ligand. By performing in vitro and in vivo experiments, they show that the nanoparticle-coupled drug enters the tumor cell in a CXCR4-dependent fashion, accumulates in the cytoplasm, and elicits a stronger cytotoxicity than the particle-free drug. Importantly, treatment of tumor xeno-engrafted mice with T22-GFP-H6-FdU effectively reduced CXCR4⁺ metastatic tumor foci in a 48 hrs time window, while normal tissue cells were not affected by the drug. Interestingly, CXCR4⁺ cells recurred 72 hrs post-treatment, presumably due to the drug pharmacokinetic, but a repetitive treatment schedule kept the metastatic foci (liver and lung) re-growth at bay. Interestingly, lymph node metastasis did not benefit from T22-GFP-H6-FdU when compared

to free Oligo-FdU. The study provides strong data that support the concept of targeted drug delivery to aggressive CXCR4⁺ cancer cells. In a pre-clinical model of tumor metastatic outgrowth, the laboratory of Ramón Mangués provides very appealing data showing that T22-GFP-H6-FdU specifically elicits DNA damage and apoptosis in CXCR4⁺ cancer cells. Normal tissue is spared from this drug which fails to enter the cell in the absence of high-level cell-surface standing CXCR4 receptor.

Overall, the study provides a novel anti-metastatic treatment strategy which very likely can be translated to the clinic.

The following points should be addressed to further improve the quality of the manuscript:

Major points

1) T22-GFP-H6-FdU shows a significant effect on lung and liver metastasis formation. However, growth of the primary tumor and lymph node metastasis is largely unaffected. The authors explain this effect by different levels of cancer cell CXCR4 expression in different locations. However, according to the IHC staining provided in Figure 4D, primary tumors and LN metastasis show indeed a reasonable CXCR4 staining which gets reduced after administration of T22-GFP-H6-FdU. Since the authors also describe efficient delivery and drug-uptake by sub-cutaneously implanted cancer cells, it should be expected that these cells, after experiencing increased DNA damage and caspase-cleavage (Fig. 2D), get largely depleted from the primary tumor. The authors should therefore provide more experimental evidence to support their hypothesis regarding the differential impact of T22-GFP-H6-FdU on primary and metastatic tumor growth. Especially, the bio-distribution of the drug to the orthotopic transplantation site and LN metastasis could be indeed worse, therefore lowering the effective drug concentration when compared to the liver or lung.

We thank the Reviewer for this insight. We agree on the point that limited nanoconjugate distribution to the colonic tumor and LN Mets could reduce the anticancer effect at these sites. In fact, we have proven that CXCR4 expression level and biodistribution are linked. Thus, we previously reported that the uptake of the T22-GFP-H6 nanoparticle (used to generate the nanoconjugate tested here) is higher in the metastatic sites showing higher CXCR4 expression in their epithelial cancer cells. This way, both, nanoparticle internalization in CXCR4⁺ cancer cells, as well as tissue uptake, in LN Mets are significantly reduced as compared to internalization in CXCR4⁺ cells and tissue uptake, for instance, in peritoneal (PTN) Mets (Céspedes 2016). On this basis, the nanoconjugate may induce a level of DNA damage and apoptosis, and a reduction in the CXCR4⁺ cell fraction remaining at the end of a repeated dose treatment, significantly lower in LN

Mets than in LV Mets, LG Mets or PTN Mets. Consistently, the nanoconjugate achieves a significantly higher blockade of Mets foci development in LV, LG and PTN than free-FdU. Nevertheless, the nanoconjugate may still induce some DNA damage and caspase-cleavage in LN Mets since it reduces the number of metastases as compared to buffer-treated animals; however, its antimetastatic effect does not differ from that achieved by free FdU at this site.

2) Related to point 1, the possibility exists that CXCR4 positive cells, which due to the obvious tumor heterogeneity (CXCR4⁺ and CXCR4⁻ cells) represent only a fraction of the tumor, are dispensable for primary tumor and LN metastasis growth. Still, these cells might be key to induce and maintain metastatic tumor growth in the lung or liver. A similar phenotype has been recently described by the laboratory of Frederic Sauvage, showing that depletion of LGR5-expressing CRC stem cells prevents the formation and growth of liver metastasis while primary tumor growth at the orthotopic transplantation site remained unaffected (FS e Melo 2017, Nature, 2017). A study by Weidong Wu et al. (Oncotarget, 2016 Dec6) shows that LGR5/CXCR4 double positive CRC cells show the highest capacity of tumor re-formation in serial transplantation experiments. Hence, T22-GFP-H6-FdU administration might indeed kill CRC stem cells which possess highest tumor/metastasis re-formation capacity. To further clarify this question, the authors should analyse as to whether treatment with T22-GFP-H6-FdU reduces overall LGR5 expression in the primary tumor orthotopic implant. It would be also very interesting to address if primary tumor cells of T22-GFP-H6-FdU treated animals possess a lower tumor-re-initiation capacity. This could be easily addressed by assessing the tumor-organoid or spheroid formation capacity of T22-GFP-H6-FdU-treated (CXCR4 low) vs non-treated (CXCR4 high) primary tumor cells.

We agree with the Reviewer's statement that the effect of CXCR4⁻ cells on cancer/metastasis growth could differ among cancer sites, being relevant in LV Mets, LG Mets and PTN Mets, but having low or no effect on LN Mets or Colonic/Primary tumor (PT) growth.

Regarding tumor re-initiation capacity, we have treated SW1417-derived SC tumors with T22-GFP-H6-FdU, Free-FdU or Buffer and found, as expected, that spheroid formation capacity was significantly lower in T22-GFP-H6-FdU than in free-FdU or Buffer as measured by bioluminescence emission and spheroid count (novel Fig. 4C-D).

Similarly, T22-GFP-H6-FdU-treated M5 SC tumors yielded a significantly lower spheroid formation than free-FdU or Buffer-treated tumors (Fig.5A-B). Most importantly, implantation of disaggregated cells from M5 SC tumors treated with T22-GFP-H6-FdU in recipient tumor-free mice yielded lower number and size of new tumors (lower re-initiation capacity) than free-FdU or Buffer-treated-tumors (Fig. 5C-D).

Regarding LGR5 expression in the orthotopic implant, surprisingly we could not detect this protein in M5 or SW1417-derived primary tumors, using two different anti-Lgr5 antibodies. However, Lgr5 was clearly present at the bottom of the crypts in the normal mouse intestine. We did not detect, either, Lgr5 in SW1417 cells in culture. (Please, see Figure for Referee 1 for this in vivo and in vitro results). A possible interpretation of these findings could be that a marker different from Lgr5, which associates also with CSC function (e.g. CD133), could be co-expressed with CXCR4 for the cells to display a MetSCs phenotype in our CRC models.

In this regard, at least ten different proteins have been proposed as markers for CRC CSCs; however, there is no consensus on which single marker, or combined markers, best determine tumor initiation, chemo-resistance or metastatic capacity in CRC (Munro 2017). Despite CD133, LGR5 and CXCR4 show all prognostic value in CRC, they display different functions. Whereas CXCR4⁺ cancer cells associate with cell trafficking and metastasis development, CD133 or Lgr5 associate instead with clonogenicity, tumorigenesis and resistance, rather than with trafficking and metastasis. Moreover, CRC cells co-expressing CXCR4 and CD133 are highly metastatic (Zhang 2012; Li 2015), as it has been described for CXCR4 and Lgr5 co-expressing cells (Wu W 2016; de Sousa e Melo 2017)

In addition, CSCs plasticity makes it complex the dissection of the molecular and functional properties of the proteins involved in the cancer stem cell phenotype. Thus, it has been proven that CSCs undergo dynamic, reversible and environment-dependent changes that concomitantly switch CSC makers. In this venue, chemotherapy treatment could enrich for CSCs, not only by selecting therapeutic resistant CSCs, but also by inducing CSC properties in non-CSCs. For instance LGR5+

tumor cell depletion blocks tumor growth, which after treatment discontinuation is followed by interconversion of Lgr5⁻ to Lgr5⁺ tumor cells, which triggers rapid tumor re-growth (Yoshida 2016, Dieter 2017, Battle 2017). Similarly, after chemotherapy, Lgr5⁺ cells can interconvert to Lgr5⁻ cells, which express CD133, and are also capable of tumor reconstitution (Kobayashi 2012).

Regarding this issue, it is interesting our novel observation suggesting that the T22-GFP-H6-FdU antimetastatic effect could, at least partially, be explained by blocking tumor cell dissemination already in the primary tumor. Thus, we found a significant reduction of intravasated CXCR4⁺ tumor emboli, in the peri-tumoral area of the M5-derived orthotopic primary tumors, as compared to free-FdU or buffer-treated mice (please, see new Fig. 5E-G), meaning that the blockade of CXCR4⁺ cancer cell trafficking (intravasation in peritumoral vessels) at the primary tumor may constitute an important component of T22-GFP-H6-FdU-induced antimetastatic effect.

Thus, Lgr5 may regulate different functions than CXCR4 regarding stemness and metastatic dissemination and growth. In Melo's article, Lgr5 ablation blocks the growth and maintenance of liver Mets but does not block primary tumor invasion (which could depend on CXCR4). In contrast, CXCR4 appears critical for cell trafficking, invasion, intravasation and metastasis (references 40-45 of our previous manuscript version), a function consistent with our finding that T22-GFP-H6-FdU blocks CXCR4⁺ tumor emboli intravasation.

Referee #2 (Comments on Novelty/Model System for Author):

To my view the manuscript focuses in a very relevant issue and should provide relevant information about cancer therapy, however the work needs to be technically improved.

We thank the Reviewer's suggestions, which we have strictly followed in answering the points below. We believe that the novel generated data have strengthened the conclusions put forward in the new manuscript version. We have now included the missing information and carefully described the number of assessed mice, samples or replicates used in Material and Methods and Figure Legends, and gave quantitative data on each assay in the Results Section, besides describing the statistical test used to compare groups.

Referee #2 (Remarks for Author):

This work focuses in a very interesting subject that is the possibility of specifically targeting/killing metastasis-initiating cells for anti-cancer therapy. Authors use a conjugate compound that includes a region that binds the CXCR4 receptor that relates with metastatic capacity in several models and a cytotoxic agent that induces DNA damage. Ideally, this compound will target metastatic cells leading to improved therapeutic activity of the toxic drug. However, different technical issues together with the lack of details about procedures, quantifications or the number of replicates performed for each experiment strongly weaken the solidity of the results shown.

Comments and suggestions to the main concerns:

1-Cxcr4 and GFP detection in IC needs to be done in double IF (fluorescent IHC) to show that GFP is specifically internalized in the CXCR4⁺ population of the tumor. Quantification of different tumor areas and animals is required.

Thank you. We have now performed double immune-fluorescent assays to show that the GFP domain of the nanoconjugate is specifically internalized in SW1417 CRC cells that overexpress the CXCR4 in their membrane, since GFP and CXCR4 co-localize in these cells (please, see Fig. 2C of the revised manuscript).

2- In Figure 3A the number of gammaH2A⁺ cells is similar in the images corresponding to T22-GFP-H6-FdU and free FdU-treated tumors. However, in the text it is said: the number of DSBs foci in tumors was significantly higher than after free oligo-FdU (22.8 {plus minus} 1.4 versus 13.4 {plus minus} 0.7; p=0.02). If they are referring to the percent of tumor cells that are positive, the sentence needs to be reformulated. This is not foci but cells. Also, I would suggest changing this

statement to: "the number of cells (per 20X field????) containing DSBs foci in the tumors was slightly but significantly higher than after free oligo-FdU...". Again, the sentence needs to include details on how these numbers were obtained.

The Reviewer is right in that we incorrectly formulated both sentences. What we counted was the number of cells containing DSBs foci rather than the number of DSBs foci induced by the nanoconjugate or free FdU. Thus, we have now substituted the term "foci" by "cells" when describing these data in Fig. 3 and the Results sections, and give detailed procedures in Material and Methods and in the Figure Legend.

3- In 3A and 3B including a bigger area of the tumor (in addition to the detail) will help to interpret the results, since small areas can be deliberately selected to illustrate any conclusion. Moreover, selective elimination of CXCR4+ cells in the different treatments and at the different periods of treatment has to be quantified from various animals in each group (IHC, flow-cytometry...).

We have included bigger tumor areas in these graphics. We have also quantitated Hoechst-stained and Immunohistochemistry stained samples of caspase-3 and H2AX-gamma (novel Fig3), and proteolyzed PARP (Suppl. Fig. 4) and the percent of CXCR4+ cells remaining at different time after treatment (novel Figure 4A-B).

4- An important question that should be tested is whether tumors lacking CRC4 are capable to metastasize (i.e. tumors treated with T22-GFP-H6-FdU for 24-48 hours) and whether CXCR4+ cells are responsible for metastasis in this particular model. Can authors compare the percent of CXCR4+ cells in the bulk of the tumor, the invasive areas and the metastasis at early stages of invasion?

Thank you for suggesting this interesting exploration. We have performed new in vivo experiments at early time points to tackle the issue. We have measured the percent of CXCR4+ intravasated tumor emboli in the peri-tumoral area in M5-derived orthotopic primary tumors. We found that the intravasated tumor emboli were significantly reduced in T22-GFP-H6-FdU as compared to free-FdU or Buffer (please, see new Fig. 5E-G). We interpret this result as an indication that the antimetastatic effect induced by T22-GFP-H6-FdU starts, and may be in part explained, by its blockade of CXCR4+ cancer cell trafficking in primary tumor (intravasation in peritumoral vessels), which could subsequently reduce tumor dissemination.

5-In 3B the quality of the images of basal and treated tumors is totally different. Is there any reason for this heterogeneity?

Thank you for your comment. The differences in image quality were probably due to different time exposure and microscope set-up. To solve this problem, we have now taken new photographs, and use the same timing and set-up, for all compared samples. We hope the Reviewer finds now the images included in previous Fig. 3B (novel Fig. 4B in the revised manuscript) comparable among groups and of sufficient quality for publication.

6-In 4B, graphs are randomly labeled and it seems that the effect of the FdU conjugate is only compared with controls but not with the free FdU-treated animals, although it is not indicated. Also, deviations are lacking. In addition, the signal associated to the metastatic component in the different animals (reflecting metastatic load) and treatments need to be shown, not just the number of foci. The same criticism applies to 4C and in this case it is unclear how the percent of remaining CXCR4+ cells in the metastasis have been calculated. Images supporting these results have to be included in the main figure.

A comparison of the metastatic load among groups is exhaustively described in Table 1 of the manuscript for both SW1417 and M5-derived models. Data in this Table describe two different components regarding metastatic load:

1) T22-GFP-H6-FdU treatment induces a significant reduction of metastatic load, which includes a significant increase in the percent of animals completely free of metastases at the end of treatment (83% Mets-free in liver (LV), lung (LG) and peritoneum (PTN) sites in the SW1417 model (achieving, therefore a complete elimination of CXCR4+ cancer cells at these sites), as compared to only 27-64% range for Mets-free animals after free-FdU or Buffer treatment. This nanoconjugate

induces also a significant reduction in metastatic load in the M5 model (Mets-free mice: 38-63% in T22-GFP-H6-FdU vs. 0-30% in free-FdU o Buffer).

2) An additional measure of T22-GFP-H6-FdU-induced reduction in metastatic load is the significant reduction in size and number of metastases in Mets positive mice in LV, LG and PTN (Mets+).

We now emphasize the combined effects of the nanoconjugate on Mets-free mice (in which a complete elimination of CXCR4⁺ cancer cells occur) and Mets-positive mice (in which only a partial reduction in CXCR4⁺ CCF happens) in the Results section and comment on its consequent reduction in metastatic load in the Discussion. We have also included an additional Figure showing the metastatic pattern observed for all evaluated SW1417 and M5 models (novel Suppl. Fig. 9).

Our objective in Fig. 4 (Fig. 6 in this new version) for the M5 model was to present graphically the main findings described in Table 1, such as the reduction in mean foci number (novel Fig. 6A) as well as the percent of CXCR4⁺ cells remaining at the metastatic tissues (CXCR4⁺ CCF) at the end of treatment (novel Fig. 6B). We have also performed the same analyses for the SW1417 model (novel Suppl. Fig. 8). In calculating the remaining CXCR4⁺ CCF, for both models, we only included Mets+ animals.

Thus, it is important here to emphasize that the reduction in CXCR4⁺ CCF measured only in Mets+ mice, although significant is an underestimation of the real reduction in CXCR4⁺ CCF is since a majority of mice were Mets-free (undergoing, therefore, complete CXCR4⁺ cancer cell elimination). Moreover, to improve the whole Figure, we have now changed the graphics depicting and statistically comparing the mean (\pm SE) number of foci per metastatic site between groups, instead of its total number (novel Fig. 6A).

In addition, we decided to delete the results on bioluminescence emission by SW1417 tumor and metastases, so that novel Fig. 6 describes now only results obtained in the patient-derived M5 model. We maintained the panel that describes IHC data on tumor and metastases in the M5 model, which we have improved by introducing representative images of the calculated mean percent of CXCR4 expression remaining in Mets+ mice, at the different sites, at the end of repeated-dose treatment (Fig. 6B). Similar results to those described in Fig. 6 for the M5 model are depicted in Suppl. Fig. 8 for the SW1417 metastatic model. All procedures to obtain these data are described in Material and Methods and the corresponding Figure Legends

7- In 4D, images are very small specially when compared with previous images. This is even worst in the case of 5B where DNA damage cannot be evaluated at all. In addition, 5B is labeled as anti-gamma-H2AX (5h) what suggest that animals were treated for 5h with this antibody. If 5h is the period of treatment with the therapeutic compounds, this needs to be better indicated.

We have changed the graphics in previous Fig. 4D (now Fig. 6C) including higher magnification images so that the percent of remaining CXCR4⁺ cells at the end of treatment could be better assessed. We have also changed to higher magnification DNA damage in previous Fig. 5B (Fig. 7B in this new version) to improve the comparison among groups, now indicating that mice were treated with a single dose of the nanoconjugate, free-FdU or Buffer, followed by mice sacrifice 5h hours later (at the DNA damage peak) to assess genotoxic damage. The procedure to measure and evaluate the differences in CXCR4 or gamma-H2AX among groups is described in detail in the Methods Section while the number of assessed mice and samples are given in Figure Legends.

8- Flow cytometry analysis of DNA damage and apoptosis in the CXCR4⁺ and CXCR4⁻ populations of tumors exposed to different treatments (i.e. in 3A and 4D) will help to support the main message of the manuscript.

Thank you for your comment. Flow cytometry measurement of DNA damage cannot be performed because nanoconjugate internalization induces CXCR4 downregulation in treated cells; that is, the receptor localizes in the cytosol so that it is no longer present in the cell membrane. Thus, cells cannot be sorted out by flow cytometry based on CXCR4 membrane expression. Instead, we have measured in SW1417-derived subcutaneous tumors, 30 min, 1h, and 5 hours after a 100ug single dose of T22-GFP-H6-FdU treatment, and found an increase in DNA damage, being DSBs measured by gamma-H2AX (Fig. 3A-B) or proteolyzed PARP(Suppl. Fig. 4), as well as an increase in caspase-3 activation and apoptosis (Fig. 3A-B), in tumor tissue, followed by the reduction in the percent of CXCR4 cancer cells 24-48h after treatment, as compared to free-FdU or Buffer (Fig. 4A-B).

9- It should be specifically mentioned or tested whether CXCR4 is expressed in any particular tissue of the body and the possible impact that the conjugate can exert in this tissue (if any).

We know that the expression of CXCR4 in normal epithelial cells is absent or very low. Consistently, we did not observe DNA damage or apoptosis, for instance, in kidney epithelial cells (previous Fig. 5B (Fig. 7B in the revised version)). Among normal tissues, CXCR4 expression is, however, high in cells of the hematopoietic system (<https://www.proteinatlas.org/ENSG00000121966-CXCR4/tissue>), being the highest in the bone marrow. Nevertheless, the levels of CXCR4 membrane expression in metastatic cancer cells is significantly (10-20 fold) higher than in CXCR4+ normal tissues, and specifically higher than in bone marrow (Kim 2005, Kim 2006, Schimanski 2005). Consequently, there is a wide margin of membrane overexpression in cancer cells as compared to normal cells, which we exploit to render the nanoconjugate capable of selectively internalizing in cancer cells, while having a negligible distribution to normal cells. To prove this point, we evaluated DNA damage and apoptosis induced by the nanoconjugate in bone marrow. The results are depicted in new Fig. 7B. We found lower DNA damage (number of cells positive for DSBs) in T22-GFP-H6-FdU-treated in normal bone marrow than in free-oligo-FdU-treated mice. Moreover, we also observed absence of apoptosis induction and lack of histological alterations in normal bone marrow in all studied groups (Fig. 7C), as compared to cancer tissue, where we observed high apoptosis histologically and after Hoechst and staining (Fig. 3A). Thus, the nanoconjugate achieves a potent antimetastatic effect without bone marrow damage or toxicity.

In general, results are shown in a very descriptive manner and even when quantifications are included it is impossible to know how they have been obtained. The number of animals used in the different experiments is not mentioned what difficult obtaining definitive conclusion. Examples are found in 2C, 3A, 3B, 5B, but all along the manuscript.

In the revised version of the manuscript, we have carefully checked the Results, Figure Legends and Materials and Methods Sections to add the missing information, so that we ensured to give a detailed description of the procedure used to quantitate each of the evaluated parameters (number of animals, tissue samples or sections, and replicates analyzed).

Referee #3 (Comments on Novelty/Model System for Author):

No endogenous expression levels of the target

Referee #3 (Remarks for Author):

Selective depletion of metastatic stem cells as therapy for human colorectal cancer María Virtudes Céspedes et al.

Summary:

One of the major concerns in the field of colorectal cancer therapy is the massive metastatic spread of the tumors. Metastatic stem cells (MetSCs) form a subset of cancer stem cells that facilitates dissemination of cancer cells, their trafficking and eventually the re-growth of tumor cells away from the primary site. Targeting the MetSCs using nanoparticle based drug delivery is one such approach which is currently widely explored in the cancer field. Cancer cell CXCR4 receptor overexpression has been associated with metastatic properties, tumor growth and poor patient prognosis. Using a drug-nanoconjugate T22-GFP-H6-FdU, specifically targeting CXCR4+ cancer cells, Céspedes et al. here describe the method to selectively target these cells and their elimination leading to antimetastatic effects. The authors study the effects and biodistribution of the nanoconjugate in CRC cell line and patient derived model system. The authors demonstrate a potent and site-dependent metastasis prevention using the nanoconjugate targeting CXCR4+ cells.

Major Points:

- The authors make use of the CXCR4 overexpressing CRC cell line model (CXCR4+ SW1417) to study the internalization of the nanoconjugate. However, they do not address this with respect to cells expressing normal levels of CXCR4 receptor. A cell line with normal expression of CXCR4

should be used as a control. Also, the amount of CXCL12, the ligand for CXCR4⁺ must be evaluated. The CXCR4/CXCL12 axis is associated with various stages of tumor metastasis. The expression levels of CXCR4 and CXCL12 (with qPCR or SDS PAGE) in the CXCR4⁺ SW1417 cell line should be addressed before and after the internalization of the nanoconjugate.

Thank you for your comment. We do not have available a normal cell line that could maintain *in vitro* the low CXCR4 levels observed in some normal cells *in vivo*. Thus, we cannot compare the nanoconjugate effect between a normal cell line and the SW1417 CXCR4⁺ CRC cell line.

Nevertheless, we have now determined the levels of CXCR4 in the SW1417 cell line, by FACS and IHC, and showed that it displays high and constitutive CXCR4 expression (Fig. 1D). In contrast, in normal cells CXCR4 is not constitutively expressed. Thus, whereas normal cells maintain CXCR4 expression *in vivo*, they downregulate this receptor as soon as they are placed in culture (e.g. leukocytes (please see reference by Nieto 2012)). This effect occurs most likely because the maintenance of CXCR4 membrane expression and trafficking functions (e.g. in mature hematopoietic cells) need a CXCL12 gradient that only occurs *in vivo*, which allows their migration towards the diseased tissues (e.g. inflammation (Domanska 2013)). Therefore, regulation of CXCR4 differs between normal and transformed cells. Many oncogenic mutations (e.g. EGFR, Ras, PI3K, p53 mutations,...) that transform epithelial cells or trigger their metastatic progression lead to constitutive CXCR4 overexpression (Pore 2006, Domanska 2013).

Despite we could not compare the nanoconjugate effect between transformed and normal cells, we have previously reported that therapeutic nanoparticles targeting CXCR4 do not internalize nor induce cell death in established cancer cell lines that lack or have very low CXCR4 expression (Sanchez-Garcia 2018)

To solve the possible toxicity issue of the nanoconjugate on normal cells, we performed *in vivo* experiments devoted to measure its biodistribution and internalization in tumors (Fig. 2) and normal tissues, measuring also DNA damage and histological alterations in parallel in the same mice (novel Fig. 7). We found that T22-GFP-H6-FdU did not induce any toxicity in normal bone marrow (the normal tissue with highest CXCR4 level). In fact, it showed a lower number of cells stained by double strand breaks (DSBs) in DNA (genotoxic damage) than free oligo-FdU-treated animals; showing also absence of histological alterations in this tissue (Fig. 7B-C). Thus, it is unlikely that in tissues with lower expression or no CXCR4 expression the nanoconjugate could induce toxicity (e.g. we have demonstrated absence of toxicity in kidney (Fig. 7B)). In addition, we did not observe any histological alteration in any of the evaluated normal tissues (brain, lung, heart, liver, kidney, bone marrow)(Fig. 7C).

Following the Reviewer's recommendation, we have also measured the levels of CXCL-12 (SDF-1 α) in the SW1417 cancer cell line, by ELISA, finding that it does not express CXCL-12 (Fig. 1E). Consistently, CXCL-12 is rarely expressed in cancer cells, being instead typically secreted by activated fibroblasts in the metastatic tissues, helping CXCR4⁺ cancer cells to migrate directionally and colonize the organ (Domanska 2013).

- A well characterized role of CXCR4 is in activation of MAPK/ERK and PI3K/AKT signaling. The authors show the selective targeting of the CXCR4⁺ cells *in vitro* and *in vivo* but do not address the physiological effect on the system. Whether the targeted killing of CXCR4⁺ cells lead to suppression of downstream signaling of CXCR4 to have an effect on tumor growth needs to be evaluated.

We believe that the inhibition of the MAPK/ERK and PI3K/AKT signaling pathways, which are activated downstream of the CXCR4 receptor, barely occur after T22-GFP-H6-FdU treatment, because of the existence of an additional activity that has a dominant effect on proliferative blockade and cell death induction.

Thus, T22-GFP-H6-FdU nanoconjugate is likely to activate two simultaneous mechanisms that induce cell death of CXCR4⁺ cancer cells. On the one hand, the T22 ligand (included in the nanoconjugate) may act as a CXCR4 antagonist and trigger the inhibition of CXCR4 through the indicated pathways. The second cell death pathway is triggered by the Floxuridine, which the nanoconjugate releases in the cell cytosol, that leads to DNA damage in the nucleus.

We now know that DNA damage (anti-PARP and gamma-H2AX) induced by the delivered drug (FdU) peaks at 1 hour (novel Suppl. Fig 4), whereas caspase-3 activation peaks at 5 hours (Fig. 3A)

in tumor tissue, and triggers apoptosis that peaks at 24h (Fig. 3B). This effects occur earlier than the time taken by the inhibition of CXCR4 downstream signaling to induce cell death (at least 48 hours (Peng 2016)). Thus, FdU-induced DNA damage and apoptosis activation is likely to be dominant over CXCR4 downstream signaling blockade. At the time at which we could measure the inhibition of CXCR4 downstream pathways by T22, cell death has been already induced by FdU.

This effect is similar to what has been reported for two drugs used to treat breast cancer: trastuzumab (an unconjugated antibody that blocks Erb2 downstream signaling) and the antibody-drug-conjugate Trastuzumab-emtamsine (T-D1). The TD-1 ADC internalizes in target Erb2 breast cancer cells and induces their death because of its intracellular release of the cytotoxic emtamsine. TD-1-induced cell death occurs at shorter times, at much lower concentrations, and using a cytotoxic mechanism that differs from trastuzumab-induced cell death. Moreover, breast cancer cells resistant to trastuzumab, are sensitive to TD-1, both *in vitro* and *in vivo* (Lewis Phillips 2008; Barok 2011).

- The authors show the selective tumor biodistribution and internalization of the nanoconjugate in the CXCR4⁺ SW1417 CRC mouse model (Figure 2), however, they do not show the uptake in a non-tumor tissue from the same model. The CXCR4 is expressed in normal cells as well. They do address the lack of T22-GFP-H6-FdU accumulation in normal tissues in Figure 5, however, it's unclear which mouse model was used for Figure 5. The tumor vs normal tissue should be addressed in the same figure for the smooth flow of the paper.

Apologies for our lack of precision in presenting the data in these two Figures. They are indeed generated in the same experiment and using the same model. Whereas Fig. 2 shows selective tumor biodistribution and internalization of the nanoconjugate in the CXCR4⁺ SW1417 subcutaneous CRC mouse model, Fig. 7A (previous Fig. 5A) depicts normal organ accumulation of the nanoconjugate in the same experiment. Thus, there is a lack of nanoconjugate uptake, as measured by fluorescence emission, in non-CXCR4 expressing organs (brain, lung, heart), in CXCR4 expressing normal organs (spleen or bone marrow) and also lack of accumulation in organs in which nanoparticles usually accumulate (kidney or liver (only transient detection at 5h and undetectable at 24h)). These findings were also associated with lack of histological alterations in all studied normal organs (Fig. 7B-C).

In addition, we performed a study of toxicity in the M5 and SW1417-derive orthotopic model at the end of the repeated dose treatment and found the same result: absence of histological alterations in CXCR4⁺ and CXCR4⁻ normal organs, which associate with lack of body weight lost in animals after treatment. The new data are described in the novel Suppl. Fig. 9.

- The authors show the ability of nanoconjugate to induce double strand breaks (DSBs) to show its capacity to release FdU in target cells to reach nucleus and induce DNA damage (Figure 3A). They look at the foci 5 hours after the treatment. However, by that time DNA damage response (DDR) will also be expected to happen. It will be good to see the foci status at an earlier time point. It's difficult to separate DSB from DDR but a time-series can be performed to state whether it is DSBs or DDR. The localization with other repair proteins can also clear between the two.

Following the Reviewer's suggestion, we have now performed a new *in vivo* experiment to assess by IHC the occurrence of DDR single strand-breaks (SSBs), measured with an anti-PARP antibody (Suppl. Fig. 4B-C), and of DSBs, measured with an anti gamma-H2AX antibody (Fig. 3A-B and Suppl. Fig. 4A) at earlier time points (30, 1h and 5h), using the SW1417 subcutaneous tumors, in order to determine which of the two processes occurs earlier). We have found that T22-GFP-H6-FdU induces SSBs already at 30 min, whereas DSBs is not induced until 1 hour; thus, SSBs appears to occur before DSBs in our model, nevertheless, both lesions peak at 1 hour. It is important here to indicate that the protein-based nanoconjugate needs to be proteolyzed for the conjugated FdU to be released, the timing for SSBs or DSBs induction could be longer than that expected for a freely diffusible genotoxic low MW drug.

- Figure 4 was not easy to follow. The authors need to rewrite the description and arrange the figure panels more clearly. They state on page 8, top line, the histological evaluation of LV, LG..... was done for Met⁺ mice however in Figure 4 we do not see the representative histology sections. Figure 4B shows the quantification of SW1417 and M5 model but panel A only shows the mets representation from SW1417 model. Also, the switch between their mouse models is random. The

writing needs to be clearer on which animal model they refer to. Also, the quantification shown in Figure 4C belongs to panel D but the title to the two panels is misleading. Both are M5-patient derived model.

Sorry for the inconvenience. We have reorganized all Figures to show in a particular Figure only data generated in the SW1417 model or in the M5 model, not both. Thus, we have deleted panel A from previous Fig. 4 (novel Fig. 6 in the revised manuscript) to show only data on the M5 model. Moreover, we have improved the images representing differences among sites in mean percent of CXCR4 expression, as measured by IHC, in the M5 model, at the end of treatment (novel Fig. 6C)

Regarding the orthotopic SW1417 model, we have now introduced Suppl. Fig. 8 that describes all results obtained in this model, including metastatic foci number and the percent of CXCR4+ expression, at the end of treatment, at the different sites.

Finally, in the novel Suppl. Fig. 9, we are depicting representative pictures of the histology in primary tumors and metastatic foci and also of normal tissues at the end of treatment with either T22-GFP-H6-FdU, free-FdU or Buffer in the M5-derived orthotopic model and in the SW1417-derived orthotopic models.

- Drug leakage is a common problem with nanoparticle derived drug targeting. The authors do not address on how much drug is actually delivered to the tumor cell.

Drug leakage during circulation in the bloodstream after i.v. administration does not happen when using an uncleavable linker, such as the one we here used (4-maleimido hexanoic acid *N*-hydroxy-succinimide ester, MHHS) to generate the T22-GFP-H6-FdU nanoconjugate. This has been extensively described for the production of other protein-based drugs such as the antibody-drug conjugates (ADCs) (McCombs 2015), and applies also to the generation of protein-based nanoconjugates, as we did. This approach does not represent a problem for the release of the drug after the intracellular uptake of the nanoparticle or ADC since they undergo complete proteolysis (McCombs 2015), a finding consistent with the DNA damage and the induction of apoptosis induced by T22-GFP-H6-FdU after its internalization in target CXCR4⁺ cancer cells, we are reporting.

Minor Points:

- The fourth line under the heading "T22-GFP-H6-FdU internalization, CXCR4 specificity and cytotoxicity in CXCR4+ cells in vitro" has a typo with SW141-luc cells (should be SW1417-luc cells).

Corrected

- The authors do not describe the methodology of the H2AX IHC but only refer to the articles. However, the articles cited 'Kuo & Yang, 2008; Podhorecka et al, 2010, on page 5 are reviews and give no details of the method in itself. The authors need to add the details of antibody dilution, incubation time to the method section.

Done

- Based on the histology sections from various tissues in different mouse models, the authors present the quantification of the number of foci in Figure S5B. However, there are no histology representative pictures to verify that. A panel of H&E stained section is present in Fig. S4 but it is not clear if they are representative of the same quantification in Fig. S5B. It will be easy to rearrange the sub-figures accordingly.

Done. Novel Suppl. Fig. 9 incorporates all missing data on histology observed in primary tumors and metastatic foci and also of normal tissues at the end of treatment with T22-GFP-H6-FdU, free-FdU or Buffer in the M5-derived orthotopic model and in the SW1417-derived orthotopic model.

References for Reviewers

Batlle E, Clevers H. Cancer stem cells revisited. *Nat Med.* 2017 Oct 6;23(10):1124-1134

- Barok M et al. Trastuzumab-DM1 causes tumour growth inhibition by mitotic catastrophe in trastuzumab-resistant breast cancer cells in vivo. *Breast Cancer Res.* 2011 Apr 21;13(2):R46.
- Cespedes MV et al. Cancer-specific uptake of a liganded protein nanocarrier targeting aggressive CXCR4+ colorectal cancer models. *Nanomed Nanotechnol Biol Med* 2016, 12(7):1987-96
- de Sousa e Melo F et al. A distinct role for Lgr5+ stem cells in primary and metastatic colon cancer. *Nature.* 2017 Mar 29;543(7647):676-680.
- Dieter SM et al. Colorectal cancer-initiating cells caught in the act. *EMBO Mol Med.* 2017 Jul;9(7):856-858.
- Domanska UM et al. A review on CXCR4/CXCL12 axis in oncology: no place to hide. *Eur J Cancer.* 2013 Jan;49(1):219-30.
- Kim J et al. Chemokine receptor CXCR4 expression in patients with melanoma and colorectal cancer liver metastases and the association with disease outcome. *Ann Surg* 2006;244:113-20.
- Kim J et al. Chemokine receptor CXCR4 expression in colorectal cancer patients increases the risk for recurrence and for poor survival. *J Clin Oncol* 2005;23:2744-53.
- Kobayashi S et al. LGR5-positive colon cancer stem cells interconvert with drug-resistant LGR5-negative cells and are capable of tumor reconstitution. *Stem Cells.* 2012 Dec;30(12):2631-44.
- Lewis Phillips GD et al. Targeting HER2-positive breast cancer with trastuzumab-DM1, an antibody-cytotoxic drug conjugate. *Cancer Res.* 2008 Nov 15;68(22):9280-90
- Li XF et al. Effect of CXCR4 and CD133 co-expression on the prognosis of patients with stage II~III colon cancer. *Asian Pac J Cancer Prev.* 2015;16(3):1073-6.
- McCombs JR, Owen SC. Antibody drug conjugates: design and selection of linker, payload and conjugation chemistry. *AAPS J.* 2015 Mar;17(2):339-51.
- Munro MJ et al. Cancer stem cells in colorectal cancer: a review. *J Clin Pathol.* 2018 Feb;71(2):110-116.
- Nieto JC et al. Selective loss of chemokine receptor expression on leukocytes after cell isolation. *PLoS One.* 2012;7(3):e31297.
- Peng SB et al. Inhibition of CXCR4 by LY2624587, a Fully Humanized Anti-CXCR4 Antibody Induces Apoptosis of Hematologic Malignancies. *PLoS One.* 2016 Mar 8;11(3):e0150585.
- Pore N et al. The chemokine receptor CXCR4: a homing device for hypoxic cancer cells? *Cancer Biol Ther.* 2006 Nov;5(11):1563-5
- Qi J et al. In vivo fate of lipid-based nanoparticles. *Drug Discov Today* 2017, 22(1):166-72
- Rosenblum D. Progress and challenges towards targeted delivery of cancer therapeutics. *Nat Commun.* 2018, 9(1):1410
- Sánchez-García L et al. Self-assembling toxin-based nano-particles as self-delivered antitumoral drugs. *J Control Release* 2018; 274:81-92
- Schimanski CC et al. Effect of chemokine receptors CXCR4 and CCR7 on the metastatic behavior of human colorectal cancer. *Clin Cancer Res* 2005;11:1743-50.
- Wu W et al. Co-expression of Lgr5 and CXCR4 characterizes cancer stem-like cells of colorectal cancer. *Oncotarget.* 2016 Dec 6;7(49):81144-81155.
- Yoshida GJ, Saya H. Therapeutic strategies targeting cancer stem cells. *Cancer Sci.* 2016 Jan;107(1):5-11.
- Zhang SS et al. CD133(+)CXCR4(+) colon cancer cells exhibit metastatic potential and predict poor prognosis of patients. *BMC Med.* 2012 Aug 7;10:85.

Figure to Referee # 1

Absence of Lgr5 expression in the SW1417 cell line. We used Immunohistochemistry to detect LGR5 expression in CXCR4⁺ SW1417 CRC cells using the anti GPCR GPR49 antibody (Abcam, Ref. ab75732; 1:100) as described by Wu W et al. *Oncotarget*. 2016;c7(49):81144-81155. We observed lack of expression of Lgr5 in this cell line before or after treatment with T22-GFP-H6-FdU. In contrast, after exposure for 48 hours to 1uM T22-GFP-H6-FdU, the number CXCR4⁺ cells was significantly reduced, as compared to equimolar free oligo-FdU or Buffer-treated cells. We obtained the same result using a different antibody (GPCR GPR49, Ref Ab219107,1:100). Scale bar, 50 μ m

Absence of Lgr5 expression in M5-derived orthotopic primary tumors. We used immunohistochemistry to detect LGR5 expression in sections of M5-derived primary colonic tumors, using the anti GPCR GPR49 antibody (Ref. ab75732; 1:100), as described by Wu W et al. *Oncotarget*. 2016 Dec 6;7(49):81144-81155. We observed lack of Lgr5 expression in all these tumors (indicated by TM in the Figure) for all of studied groups (T22-GFP-H6-FdU, free oligo-FdU or Buffer). In contrast, Lgr5 stained positive for cells located at the bottom of the crypts (indicated by black arrows) in the normal colonic mucosa (Mu), as it is expected for normal intestinal stem cells, which express Lgr5. The same results were obtained using a different antibody (GPCR GPR49, Ref. Ab219107, 1:100) and primary colonic tumor derived from the CXCR4⁺ SW1417 cell line (data not shown). Magnifications at 200x (upper panels) and x1000 (lower panels)

Thank you for the submission of your revised manuscript to EMBO Molecular Medicine. We have now received the enclosed reports from the referees that were asked to re-assess it. As you will see, reviewers 1 and 3 are now globally supportive but referee 2 remains unsatisfied. This referee has a few more concerns that must be experimentally addressed when needed. I'd like to remind you that EMBO Molecular Medicine normally invites a single round of main revision, therefore this is to be considered the last opportunity to satisfy referee 2.

In addition, should the manuscript move forward, would you please address editorial amendments. [not listed].

REFEREE REPORTS.

Referee #1 (Comments on Novelty/Model System for Author):

1. The manuscript data have been improved technically by providing detailed information on number of animals used in each experiment, procedures have been described in more detail, and statistical analyses have been revised where this was necessary.
2. A specific effect of a CXCR4 surface receptor-targeted drug on liver and lung metastatic colorectal cancer tumor cells has not been described before to our knowledge.
3. The medical impact is high since the characterized drug shows little to no side-effects on normal tissue cells. Hence, the drug might be a strong candidate for clinical trials. The model system is adequate. CRC cell line and primary CRC cells have been used. 4. Importantly, orthotopic transplantation has been performed in order to study the metastatic process of cancer progression. This method has become the state-of-the-art, and it outperforms intra tail-vein or intrasplenic cancer cell injections since it recapitulates the complete metastatic process.

Referee #1 (Remarks for Author):

In their revised manuscript, María Virtudes Céspedes and colleagues have responded to initial doubts and questions by performing new experiments, by re-structuring the manuscript for improved clarity, and by adding critical aspects to the discussion of their data. This has improved the quality of the manuscript, and it has strengthened the initially submitted data.

- The data showing that drug biodistribution correlates to the CXCR4 surface level of cancer cells explains the partially selective effect of T22-GFP-H6-FdU on the liver mets. Although the authors admit that DNA damage and apoptosis might also occur in the primary tumor after treatment, this effect might not reach the threshold to translate into tumor shrinkage. Especially the new observation made by the authors that T22-GFP-H6-FdU indeed reduces primary tumor cell dissemination e.g. to peritoneal vessels (as shown in new Fig.5E-G) provides additional mechanistic insight into how T22-GFP-H6-FdU treatment might counteract the formation of liver metastasis.

- The spheroid forming and re-implantation experiments now shown in the manuscript have proven a lower re-initiating capacity of tumor cells which had been exposed to T22-GFP-H6-FdU while exposure to free oligo-FdU was less effective in these scenarios.

- LGR5 detection in the tumor models was unfortunately not successful with the antibody and/or protocol used by the authors. Importantly, the stainings shown in figure to referee#1 do not provide any evidence of being specific. Especially, only a single cell within a normal crypt seems to stain positive and there are several signals in the sub-mucosal compartment. We admit that LGR5 staining by IHC is extremely challenging and other methods, such as in situ hybridization or a simple quantitative real-time PCR to detect LGR5 gene expression before and after treatment with T22-GFP-H6-FdU, might have been more suitable to address our question. However, I agree with the authors that other markers or marker combinations have not been established sufficiently yet in order to conclude from them the CRC cell metastatic capacity. E.g. CD133 failed to label intestinal stem cell like tumor cells in human CRC samples (Merlos et al. 2011, Cell Stem Cell) but, as cited by the authors, can specify metastatic cells in combination with CXCR4, similar to the LGR5/CXCR4 combination. Although it would be interesting to know the abundance of these other

marker proteins in CRC cells prior to and after treatment with T22-GFP-H6-FdU, we admit that these analyses might go beyond the scope of the here presented study. The observation made by the authors that drug-mediated depletion of CXCR4+ cells reduces the liver metastatic capacity suggests that CXCR4+ cells indeed represent functional cancer self-renewing cells at this metastatic site.

- Since the points made by us and, as far as we can judge on it from our point of view, by the other referees have been largely, although not completely, addressed/discussed appropriately in our opinion, we think that the current revised manuscript is suitable for publication in EMBO Molecular Medicine.

Referee #2 (Comments on Novelty/Model System for Author):

The work is interesting and relevant although I find the technical quality of several experiments insufficient to take definitive conclusions. Also, I indicate as high the medical impact but this is totally based on how well demonstrated the conclusions of the work are and whether they can be further validated in more physiological cancer models.

Referee #2 (Remarks for Author):

The manuscript has been improved following reviewers' recommendations. However, there are still some relevant issues that make the manuscript difficult to follow.

Specifically, what is more confusing to me is that free oligo-FdU is not detected in any of the experiments shown. These results could indicate less drug internalization, however I cannot see in the text or methods that the free compound is GFP-labeled. I am probably wrong but in my opinion the appropriate control for all these experiments should be the same molecule without the T22 fragment. Is this the case? If not, including GFP data from the free oligo-FdU in several panels of figure 2 is at least confusing.

Also important, in Figure 2C, CXCR4 and GFP staining are surprisingly overlapping in great contrast with the more convincing images shown in 1G where the compound is internalized and detected as intracellular dots. Do the authors interpret that the drug is persistently retained in the cell membrane bound to the receptor in the tumor cells? This result should be confirmed by including multiple controls and all single-antibody stainings, If confirmed, these results need to be discussed in detail and justified. Also a quantification of the percent of double positive and single positive (for CXCR4 and the conjugate) cells from different tumor areas would be informative.

Specific points:

In 1F it is mentioned, "AMD3100, a CXCR4 antagonist, was able to down regulate CXCR4 receptor in the membrane and completely blocked Nano-conjugate internalization (Fig 1F)." However, the decrease in membrane-exposed CXCR4 is not shown in the figure. Double detection of CXCR4 and the fluorescent compound should be included. An additional control with sh-CXCR4 treated cells would help to validate this conclusion.

What is the red staining in 1G? In the same figure, are all cells similarly internalizing FdU? If not, are the cells with higher FdU internalization inducing higher DNA damage?

In Figure 3 magnification of the different panels is clearly different, with CXCR4 panels much smaller. Sequential sections and images of comparable magnification need to be shown to confirm selectivity of the treatment. In addition, it remains unclear what γ -H2Ax stained positive cells means. Are the authors representing the number of positive cells per field with a 200X magnification? These data are only informative when referred to particular tumor areas, relative to the total number of cells per field or any type of relative quantification, by the way it is represented is empty of significance. Moreover, the cellular density in the different areas and conditions need to be considered as they can definitely modify the results obtained.

In the metastasis assays in Figure 6 it is not mentioned what the staining is, most likely CXCR4? Also it is barely explained what the images represent and how the authors have determined the

presence of tumors in the M5 models? Do they perform the analysis from the IHC images shown in Figure 6B or from H&E staining of multiple sections (that are not shown)? For me the way the analysis was done is very unclear and difficult to evaluate specially when images are not shown.

In addition, inoculating the cells in the cecum of the mice in medium greatly increases the possibility of extravasation to the peritoneum and makes the procedure totally dependent on how animals are manipulated. Also, primary high sensitivity tumors are not shown.

In general, the results section should be carefully revised for clarity since in several sections there is no clear explanation of the results obtained, what is shown in the figures and how data have been analyzed.

Referee #3 (Remarks for Author):

The manuscript by Cespedes et al. focuses on a very relevant topic of specifically targeting metastasis initiating cells in colorectal cancer and thus a potential therapeutic intervention. The authors have considerably improved the manuscript by incorporating the additional data. The writing is also much improved with the new version and also provide clear methodology for the experiments.

2nd Revision - authors' response

3rd August 2018

ANSWER TO REVIEWERS COMMENTS:

***** Reviewer's comments *****

Referee #1 (Comments on Novelty/Model System for Author):

1. The manuscript data have been improved technically by providing detailed information on number of animals used in each experiment, procedures have been described in more detail, and statistical analyses have been revised where this was necessary.
2. A specific effect of a CXCR4 surface receptor-targeted drug on liver and lung metastatic colorectal cancer tumor cells has not been described before to our knowledge.
3. The medical impact is high since the characterized drug shows little to no side-effects on normal tissue cells. Hence, the drug might be a strong candidate for clinical trials. The model system is adequate. CRC cell line and primary CRC cells have been used.
4. Importantly, orthotopic transplantation has been performed in order to study the metastatic process of cancer progression. This method has become the state-of-the-art, and it outperforms intra tail-vein or intrasplenic cancer cell injections since it recapitulates the complete metastatic process.

We thank Reviewer #1 for recognizing the novelty of our targeted drug delivery approach, as well as the achievement of a potent antimetastatic effect through the specific delivery of Floxuridine to CXCR4⁺ cancer cells leading to their selective elimination, while having negligible biodistribution and lack of toxicity in normal tissues. We also appreciate his/her statement putting up the notion that the tested nanoconjugate may be a strong candidate for clinical trials.

Referee #1 (Remarks for Author):

In their revised manuscript, María Virtudes Céspedes and colleagues have responded to initial doubts and questions by performing new experiments, by re-structuring the manuscript for improved clarity, and by adding critical aspects to the discussion of their data. This has improved the quality of the manuscript, and it has strengthened the initially submitted data.

- The data showing that drug biodistribution correlates to the CXCR4 surface level of cancer cells explains the partially selective effect of T22-GFP-H6-FdU on the liver mets. Although the authors admit that DNA damage and apoptosis might also occur in the primary tumor after treatment, this effect might not reach the threshold to translate into tumor shrinkage. Especially the new observation made by the authors that T22-GFP-H6-FdU indeed reduces primary tumor cell dissemination e.g. to peritoneal vessels (as shown in new Fig.5E-G) provides additional mechanistic insight into how T22-GFP-H6-FdU treatment might counteract the formation of liver metastasis.

Thank you for asking for additional mechanistic insights on the nanoconjugate antimetastatic effect and finding reasonable our interpretation that the high surface CXCR4 level in liver or lung metastases may explain the higher effect observed because of a higher nanoconjugate uptake at these organs, as compared to primary tumor or lymph node metastases. We also appreciate his/her support to the notion that the novel finding of a reduction in intravasated CXCR4⁺ tumor emboli induced by the nanoconjugate in peri-tumoral vessels of the primary tumor, may contribute to the observed antimetastatic effect at the different sites.

- The spheroid forming and re-implantation experiments now shown in the manuscript have proven a lower re-initiating capacity of tumor cells which had been exposed to T22-GFP-H6-FdU while exposure to free oligo-FdU was less effective in these scenarios.

Thank you for appreciating the additional contribution to the new manuscript version that represents the demonstration of a lower tumor re-initiation capacity after T22-GFP-H6-FdU treatment, and its consistency with the selective elimination of cancer stem cells and with the observed antimetastatic activity induced by the nanoconjugate.

- LGR5 detection in the tumor models was unfortunately not successful with the antibody and/or protocol used by the authors. Importantly, the stainings shown in figure to referee#1 do not provide any evidence of being specific. Especially, only a single cell within a normal crypt seems to stain positive and there are several signals in the sub-mucosal compartment. We admit that LGR5 staining by IHC is extremely challenging and other methods, such as in situ hybridization or a simple quantitative real-time PCR to detect LGR5 gene expression before and after treatment with T22-GFP-H6-FdU, might have been more suitable to address our question. However, I agree with the authors that other markers or marker combinations have not been established sufficiently yet in order to conclude from them the CRC cell metastatic capacity. E.g. CD133 failed to label intestinal stem cell like tumor cells in human CRC samples (Merlos et al. 2011, Cell Stem Cell) but, as cited by the authors, can specify metastatic cells in combination with CXCR4, similar to the LGR5/CXCR4 combination. Although it would be interesting to know the abundance of these other marker proteins in CRC cells prior to and after treatment with T22-GFP-H6-FdU, we admit that these analyses might go beyond the scope of the here presented study. The observation made by the authors that drug-mediated depletion of CXCR4⁺ cells reduces the liver metastatic capacity suggests that CXCR4⁺ cells indeed represent functional cancer self-renewing cells at this metastatic site.

We recognize that the set-up of the method to measure Lgr5 level in tumors may have needed the development of an additional experimental procedure (for instance, using in situ hybridization). Thus, this is a pending issue that we would like to address in the future in our model. Nevertheless, as the Reviewer states the study of Lgr5 or other markers, additional to CXCR4, might go beyond the scope of the study. Moreover, despite recognizing that cancer stem cells expressing other markers or their combinations could be involved in the observed antimetastatic effect; the Reviewer supports the view of one of the central findings of the manuscript; that is, the elimination of CXCR4⁺ cells by the nanoconjugate validates CXCR4⁺ cancer cells as functional cancer self-renewing cells at the liver, lung and peritoneal metastatic sites.

- Since the points made by us and, as far as we can judge on it from our point of view, by the other referees have been largely, although not completely, addressed/discussed appropriately in our opinion, we think that the current revised manuscript is suitable for publication in EMBO Molecular Medicine.

We thank the Reviewer for supporting the publication of the revised manuscript in EMBO Molecular Medicine, and also for helping us to improve its structure and the technical description of the performed experiments and the statistical analysis of the obtained results. We also thank his/her recognition of the adequacy of the cancer models used in the study. Finally, we would like to appreciate the relevant contribution of the Reviewer in emphasizing the link between CXCR4 expression level, nanoconjugate biodistribution and antimetastatic effect at the different sites.

Referee #2 (Comments on Novelty/Model System for Author):

The work is interesting and relevant although I find the technical quality of several experiments insufficient to take definitive conclusions. Also, I indicate as high the medical impact but this is totally based on how well demonstrated the conclusions of the work are and whether they can be further validated in more physiological cancer models.

Referee #2 (Remarks for Author):

The manuscript has been improved following reviewers' recommendations. However, there are still some relevant issues that make the manuscript difficult to follow.

Specifically, what is more confusing to me is that free oligo-FdU is not detected in any of the experiments shown. These results could indicate less drug internalization, however I cannot see in the text or methods that the free compound is GFP-labeled. I am probably wrong but in my opinion the appropriate control for all these experiments should be the same molecule without the T22 fragment. Is this the case? If not, including GFP data from the free oligo-FdU in several panels of figure 2 is at least confusing.

We want to thank Reviewer #2 for recognizing the improvement of this new manuscript version, by incorporating the new data following the Reviewers' recommendations. We particularly appreciate his/her suggestion for new experiments that generated novel and relevant results, including the observation of a significant reduction in the percent of CXCR4⁺ intravasated tumor emboli in the peri-tumoral area of the orthotopic primary tumors treated with the nanoconjugate. We also appreciate his/her contribution to improve the description of the obtained results, the protocols and statistical analyses used, and especially in helping clarify the two components of the reduction of metastatic load observed after T22-GFP-FdU treatment. These two components were: achieving a significant increase in the percent of animals completely free of metastases (therefore reaching a complete CXCR4⁺ cancer cell elimination) and a reduction in the size and number of metastases in the rest of the animals, as compared to free oligo-FdU treated animals.

Regarding the issue of the most appropriate control to use, when testing T22-GFP-H6-FdU antimetastatic effect, we think that free oligo-FdU may be the best control because of the following reasons:

1. The novel therapeutic approach we developed mainly pursues targeted drug delivery, which aims at improving the therapeutic window of the payload drug (Das et al, 2009) by enhancing tumor uptake and therapeutic effect (by achieving selective CXCR4⁺ cancer cell killing) as well as decreasing its biodistribution to normal tissues, and therefore, greatly diminishing its toxicity. We believe that to demonstrate that we reach this goal, we should use as best control the unconjugated drug (oligo-FdU). This is a low molecular weight (MW) drug (as the cytotoxic agents used in current cancer chemotherapy are) that crosses the membrane of all cells in the body and freely diffuses through all tissues. Thus, the designed experiments tried to answer whether selective delivery of a cytotoxic drug to the cells responsible for metastasis initiation and maintenance achieves a wider therapeutic index than the direct injection of the free cytotoxic drug.

2. We believe that using GFP-H6-FdU (which lacks the T22 ligand) will not answer the question on whether a targeted drug delivery approach increases the therapeutic window as compared to the low MW drug currently used to treat cancer. Instead, it will only give information on whether the lack of the T22-ligand changes the biodistribution of the conjugate, without estimating the possible improvement afforded over currently used chemotherapeutic drugs. To support this argument, we previously reported that the GFP-H6 protein dramatically changes its biodistribution, as compared to the T22-GFP-H6 nanoparticles, by displaying a much shorter circulation time in blood (because of enhanced renal filtration), much shorter exposure time in tumor tissue and by the expected observation of its lack of internalization in CXCR4⁺ epithelial cancer cells (internalizing instead in stromal cells) (Céspedes et al, 2016).

3. Using GFP or another method to label oligo-FdU may not be useful since the generated molecule is likely to change its biodistribution and probably its therapeutic effect and toxicity on normal tissues as reported for fluorescently labeled low MW drugs (Mérián et al, 2012).

In Figure 2, we used oligo-FdU (despite not being fluorescently labeled), or Buffer, as negative controls to show that the background fluorescence in tumor tissues is sufficiently low so that we could assess the main point described in this Figure, which is whether or not the addition of oligo-FdU to the targeting fluorescent protein-nanoparticle (T22-GFP-H6) changes the nanoparticle

biodistribution in tumor tissues. We have demonstrated that the T22-GFP-H6-FdU nanoconjugate has a similar tumor uptake as T22-GFP-H6 (Fig. 2B). If the addition of the drug would have changed the conformation of the T22-GFP-H6 nanoparticle and/or blocked the targeting capacity of the novel targeted drug delivery approach the goal that we are pursuing would not have come through.

Also important, in Figure 2C, CXCR4 and GFP staining are surprisingly overlapping in great contrast with the more convincing images shown in 1G where the compound is internalized and detected as intracellular dots. Do the authors interpret that the drug is persistently retained in the cell membrane bound to the receptor in the tumor cells? This result should be confirmed by including multiple controls and all single-antibody stainings. If confirmed, these results need to be discussed in detail and justified. Also a quantification of the percent of double positive and single positive (for CXCR4 and the conjugate) cells from different tumor areas would be informative.

We believe that Figure 1G and Figure 2C are not comparable regarding CXCR4 and GFP overlapping. Figure 1G shows only T22-GFP-H6-FdU internalization in CXCR4⁺ cancer cells *in vitro*, by direct measurement of the fluorescence emitted by the GFP domain of the nanoconjugate. The red staining in this picture corresponds to plasma cell membranes stained with a red dye (CellMask™), whereas the cell nucleus was stained in blue using the Hoescht dye. CXCR4 expression was not assessed in this experiment.

In contrast, Figure 2C displays the co-localization of CXCR4 and GFP, using specific monoclonal antibodies to detect each protein, a result that specifically addressed the Reviewer's comment, formulated as a point to answer, in the previous manuscript version. In our view, the finding of CXCR4 and GFP overlapping mainly in the cell membrane would be expected since CXCR4 is mostly expressed in the membrane and because the nanoconjugate enters the cell through receptor-mediated endocytosis. The nanoconjugate binds through its T22 domain to the CXCR4 receptor located in the membrane (thus, GFP and CXCR4 overlapping is expected in the membrane (yellow staining)) and can also be found in the endocytic vesicles that are formed and traffic towards the cytosol (yellow dots). Once in the cytosol, the vesicles release its content, while the empty endocytic vesicles, containing the unbound CXCR4 receptor (red dots), are being recycled back to the membrane (an event that happens for all G-protein coupled receptors, including CXCR4 (Venkatesan 2003)). Following the Reviewer suggestion, we have now added magnified insets to the pictures depicted in Fig. 2C where it can be observed, in the same cell, green dots that correspond to nanoparticles present in the cell cytosol (most likely released by the endocytic vesicles, since they are located away from the cell membrane) and red dots of endocytic vesicles containing empty CXCR4 receptors. Besides, these new observations, we also found as expected, in the same cell, yellow dots of co-localized CXCR4 receptors and nanoconjugates mostly in the cell membrane but also as nanoconjugate-loaded endocytic vesicles. Following this Reviewer recommendation, we have also quantitated the area occupied by green dots (nanoconjugate) and red dots (endocytic vesicles with the empty CXCR4 receptor) in single cells, and obtained a mean±SE measurement, which is described within the added insets. They clearly show that in T22-GFP-H6-FdU treated tumors green dots are present, whereas in free oligo-FdU-treated tumors they are barely detectable. The justification of the graphic displayed in Fig 2C, including the insets and the described quantitation has now been introduced in the corresponding Figure Legend.

Specific points:

In 1F it is mentioned, "AMD3100, a CXCR4 antagonist, was able to down regulate CXCR4 receptor in the membrane and completely blocked Nano-conjugate internalization (Fig 1F)." However, the decrease in membrane-exposed CXCR4 is not shown in the figure. Double detection of CXCR4 and the fluorescent compound should be included. An additional control with sh-CXCR4 treated cells would help to validate this conclusion.

We did not studied CXCR4 downregulation by AMD3100 since it has been previously established by Cheng et al. (2000), who described that the exposure of CXCR4⁺ cells in culture to AMD3100 highly reduces CXCR4 surface expression, as measured by flow cytometry (Cheng et al, 2000). We have also confirmed this finding in additional CXCR4⁺ cell lines. Thus, we observed downregulation of CXCR4 expression from the cell membranes after AMD3100 treatment in CXCR4 lymphoma cell lines, which led to a block in cell migration towards a SDF-1 α gradient (Moreno et al, 2015). In addition, we have described that AMD3100 induced downregulation of CXCR4 receptor in the cell membrane blocks the internalization of a drug-loaded CXCR4-targeted nanoparticle and its

antitumor effect *in vitro* (de la Torre et al, 2015). Based on this previous work by our group and others, we believe that our argument that AMD3100-induced CXCR4 downregulation from the membrane blocks T22-GFP-H6-FdU internalization in CXCR4⁺ cells (depicted in Figure 1F) is supported. We have now added the references supporting CXCR4 downregulation after AMD3100 exposure in the results section.

What is the red staining in 1G? In the same figure, are all cells similarly internalizing FdU? If not, are the cells with higher FdU internalization inducing higher DNA damage?

The red staining in Figure 1G is CellMask (TM), a dye used in confocal microscopy to stain the plasma cell membrane, as described in Material and Methods section (page 13). We apologize since we forgot to put it in the Figure Legend; we have now corrected it. Since Floxuridine is a low MW drug that freely diffuses through cell membranes, exposure of cells to this free (unconjugated) drug is expected to achieve a similar concentration inside all cells (independently of the cell level of CXCR4 expression); thus, leading to a similar induction of DNA damage in all cell types. In contrast, the selective internalization of the nanoconjugate T22-GFP-H6-FdU in CXCR4⁺ cells leads to a high increase in antitumor activity (IC₅₀ 18.7=nM9), as compared to free Floxuridine (IC₅₀=275 nM) (Figure 1H), most likely due to a high increase in the number of Floxuridine molecules delivered, by the internalized nanoconjugate, in their cytosol and the subsequent increase in DNA damage.

In Figure 3 magnification of the different panels is clearly different, with CXCR4 panels much smaller. Sequential sections and images of comparable magnification need to be shown to confirm selectivity of the treatment. In addition, it remains unclear what γ -H2AX stained positive cells means. Are the authors representing the number of positive cells per field with a 200X magnification? These data are only informative when referred to particular tumor areas, relative to the total number of cells per field or any type of relative quantification, by the way it is represented is empty of significance. Moreover, the cellular density in the different areas and conditions need to be considered as they can definitely modify the results obtained.

Following the recommendation of the Reviewer made for the previous manuscript version, we reduced the magnification of the CXCR4 panels so that a wider view of the tumor could be assessed. We, however, did not reduce the magnification of the γ -H2AX stained panels so that the included pictures showed enough level of detail to identify the nuclear staining pattern (DNA damage) of positive cells. It was impossible for us to fit in the same Figure high and low magnification pictures for each panel because of limited space availability. The γ -H2AX stained cells are represented with tissues acquired at 400X magnification. Positive cells are defined as cells having their nucleus stained by the anti- γ -H2AX antibody (as shown in Figure 3A), that is, the cells stained for DNA damage. We used the same magnification field and counted 10 different areas for each tumor. Tumors derived from the SW1417 cell line showed similar cell densities in their viable areas; thus, in each counted field a similar number of total cells were present. Moreover, cell counting was performed by two independent investigators, who were blinded to the evaluated groups, and reached highly concordant results, a procedure that we believe validates the analyses.

In the metastasis assays in Figure 6 it is not mentioned what the staining is, most likely CXCR4? Also it is barely explained what the images represent and how the authors have determined the presence of tumors in the M5 models? Do they perform the analysis from the IHC images shown in Figure 6B or from H&E staining of multiple sections (that are not shown)? For me the way the analysis was done is very unclear and difficult to evaluate specially when images are not shown.

We apologize for not mentioning that the tissues shown were stained with an anti-CXCR4 antibody, which is now indicated in the Figure Legend. Nevertheless, the IHC CXCR4 staining could be probably deduced from the title of the graphic that reads "Remaining CXCR4⁺ cell fraction". We also apologize for not being explicit enough in the description of the procedure followed. We first identified the presence of tumor foci in all metastatic locations in H&E stained tissue sections. Following, each section containing metastatic foci was stained with the specific antibody to determine CXCR4 intensity and stained area. We have now clarified this procedure in material and methods and in the Figure Legend. Since H&E and CXCR4-stained sections could not fit in a single Figure, because of space limitation, we decided to show representative H&E stained sections of the different tissues containing metastatic foci in Appendix. Fig. S9.

In addition, inoculating the cells in the cecum of the mice in medium greatly increases the possibility of extravasation to the peritoneum and makes the procedure totally dependent on how animals are manipulated. Also, primary high sensitivity tumors are not shown.

We implant the colorectal cancer cells in the cecal wall using a micropipette that allows their careful deposition far away from the point of injection (where the micropipette tip breaks the serosa layer), so that cell reflux is unlikely. Moreover, once the cells are inoculated and the micropipette removed from the cecum wall, we use BioGlue surgical adhesive (CryoLife Inc, Kennesaw, GA, USA) to seal the entry point to ensure that none of the injected cells can reach the peritoneum through the transcelomic cavity during the injection procedure. This point has now been added in the Methods Section.

In general, the results section should be carefully revised for clarity since in several sections there is no clear explanation of the results obtained, what is shown in the figures and how data have been analyzed.

Thank you. We have now made an additional effort to make sure that the data are clearly presented in the Results section and to improve the description of the procedures, the data obtained and the statistical analyses performed in each Figure Legend.

Referee #3 (Remarks for Author):

The manuscript by Céspedes et al. focuses on a very relevant topic of specifically targeting metastasis initiating cells in colorectal cancer and thus a potential therapeutic intervention. The authors have considerably improved the manuscript by incorporating the additional data. The writing is also much improved with the new version and also provide clear methodology for the experiments.

We thank Reviewer #3 for recognizing the relevance of achieving selective targeting of metastatic stem cells, leading to a potent antimetastatic effect in colorectal cancer models, and the potential of its clinical translation. We also thank the Reviewer for his/her suggestions that led to incorporate new results, obtained in additional experiments, as well as to improve the manuscript structure, clarity and methodological detail.

We especially appreciate the Reviewer' contribution to emphasize the wide therapeutic window displayed by the nanoconjugate, which derives from its targeting capacity, which exploits the large differences in CXCR4 expression between tumor and normal tissues and also by asking to incorporate additional data to further describe the genotoxic action of the payload drug.

REFERENCES TO REVIEWERS

- Céspedes MV, Unzueta U, Álamo P, Gallardo A, Sala R, Casanova S, Pavón MA, Mangues MA, Trías M, López-Pousa A, Villaverde A, Vázquez E, Mangues R (2016) Cancer-specific uptake of a liganded protein nanocarrier targeting aggressive CXCR4+ colorectal cancer models. *Nanomedicine* 12:1987-1996
- Cheng ZJ, Zhao J, Sun Y, Hu W, Wu YL, Cen B, Wu GX, Pei G (2000) Beta-arrestin differentially regulates the chemokine receptor CXCR4-mediated signaling and receptor internalization, and this implicates multiple interaction sites between beta-arrestin and CXCR4. *J Biol Chem* 275(4):2479-2485
- Das M, Mohanty C, Sahoo SK (2009) Ligand-based targeted therapy for cancer tissue. *Expert Opin Drug Deliv* 6:285-304
- de la Torre C, Casanova I, Acosta G, Coll C, Moreno MJ, albericio F, Aznar E, Mangues R, Royo M, Sancenón F, Martínez-Mañez R (2015) Gated Mesoporous Silica Nanoparticles Using a Double-Role Circular Peptide for the Controlled and Target-Preferential Release of Doxorubicin in CXCR4-expressing Lymphoma cells. *Adv Funct Mater* 25(5):687–695

- Mérian J, Gravier J, Navarro F, Texier I (2012) Fluorescent nanoprobe dedicated to in vivo imaging: from preclinical validations to clinical translation. *Molecules* 17(5):5564-5591
- Moreno MJ, Bosch R, Dieguez-Gonzalez R, Novelli S, Mozos A, Gallardo A, Pavón MA, Céspedes MV, Grañena A, Alcoceba M, Blanco O, Gonzalez-Díaz M, Sierra J, Manges R, Casanova I (2015) CXCR4 expression enhances diffuse large B-cell lymphoma dissemination and decreases patient survival. *J Pathol* 235(3):445-455
- Venkatesan S, Rose JJ, Lodge R, Murphy PM, Foley JF (2003) Distinct mechanisms of agonist-induced endocytosis for human chemokine receptors CCR5 and CXCR4. *Mol Biol Cell* 14(8):3305-3324

Corresponding Author Name: Ramon Mangues, Ph D

Manuscript Number: EMM-2017-08772